# A New Constraint on the Physicochemical Condition of Mars Surface during the Amazonian Epoch Based on Chemical Speciation for Secondary Minerals in Martian Nakhlites

**Hiroki Suga [1,*,†], Keika Suzuki [2,†], Tomohiro Usui [3], Akira Yamaguchi [4], Oki Sekizawa [1], Kiyofumi Nitta [1], Yasuo Takeichi [5], Takuji Ohigashi [6] and Yoshio Takahashi [2,†]**

[1] Spectroscopy and Imaging Division, Japan Synchrotron Radiation Research Institute (JASRI/SPring-8), Hyogo 679-5198, Japan; sekizawa@spring8.or.jp (O.S.); nittak@spring8.or.jp (K.N.)
[2] Department of Earth and Planetary Science, Graduate School of Science, The University of Tokyo, Tokyo 113-0033, Japan; 1614085@alumni.tus.ac.jp (K.S.); yakaha@eps.s.u-tokyo.ac.jp (Y.T.)
[3] Institute of Space and Astronautical Science (ISAS), Japan Aerospace Exploration Agency (JAXA), Kanagawa 252-5210, Japan; usui.tomohiro@jaxa.jp
[4] National Institute of Polar Research, Tokyo 190-8518, Japan; yamaguch@nipr.ac.jp
[5] Institute of Materials Structure Science, High-Energy Accelerator Research Organization, Ibaraki 305-0801, Japan; yasuo.takeichi@kek.jp
[6] UVSOR Facility, Institute for Molecular Science, Okazaki 444–8585, Japan; ohigashi@ims.ac.jp
* Correspondence: hiroki-suga@spring8.or.jp; +81-791-58-0833
† These authors contributed equally to this work.

**Abstract:** Iddingsite in Martian nakhlites contains various secondary minerals that reflect water–rock interaction on Mars. However, the formation processes of secondary Fe minerals in iddingsite are unclear because they include carbonates precipitated under reductive and alkaline conditions and sulfates that are generally precipitated under oxidative and acidic conditions. Mineral types cannot coexist under equilibrium. Herein, we characterize the carbonate phase of meteorite Yamato 000593 as siderite and Mn-bearing siderite via field-emission electron probe microanalyzer (FE-EPMA). Then, we examined the distribution and speciation of trace Cr and S within the carbonates through synchrotron micro-focused X-ray fluorescence-X-ray absorption fine structure and scanning transmission X-ray microscopy (μ-XRF-XAFS/STXM) analysis to estimate the transition history of Eh-pH conditions during siderite formation to explain the coexistence of carbonate and sulfate phases in the nakhlite vein. Specifically, the distribution and speciation of S in the mesostasis and carbonate phases and the heterogeneous distribution of Mn-FeCO$_3$ incorporating Cr(III) in the carbonate constrain the Eh-pH condition. The conditions and transition of the fluid chemistry determined herein based on speciation of various elements provide a new constraint on the physicochemical condition of the water that altered the nakhlite body during the Amazonian epoch.

**Keywords:** nakhlite; μ-XRF-XAFS; chemical speciation; Mars; martian water; aqueous alteration; Y 000593; S chemistry; Mn chemistry; Cr chemistry

## 1. Introduction

Although the current Martian surface environment is cold and dry, past Martian climate was generally wetter and warmer, and surface oceans were present with liquid water supporting habitability, as recently assessed in the literature [1]. Secondary minerals (e.g., clay minerals, sulfates, and carbonates) found on the surface of the planet reflect previous and present water–rock reactions (e.g., [2,3]). In addition, a recent investigation on brecciated Martian meteorites implies that liquid water existed on Mars until about 2100 Ma during the early Amazonian epoch [4]. Nakhlite Martian meteorites contain iddingsite, which is characterized by an aqueous-altered texture. According to

isotopic chronometer analyses of K-Ar and Rb-Sr, iddingsite formed in 633 ± 23 Ma during the Amazonian epoch [5]. Therefore, investigating the vestiges of Martian aqueous events preserved in Martian meteorites as secondary minerals is important. Such analysis can reveal details of past Martian liquid water conditions (e.g., cation composition and pH) to better understand past and present Martian habitability.

Rich in clinopyroxene (augite), nakhlites are igneous rocks that formed from basaltic magma on Mars about 1.3 Ga. According to isotopic chronometer analyses of Rb-Sr, Sm-Nd [6], U-Pb, and Ar-Ar [7], the Amazonian volcanic products on Mars were located in the Tharsis and Elysium regions, part of which originated from the volcanic activity of Olympus Mons [8]. The formation mechanism of the nakhlite host rock has been related to the basaltic volcanic activity as (i) one dynamic sedimentation event of massive thick lava flow [9,10] and as (ii) successive sequential sedimentation events of the lava flow. The mineral composition ratio in each lava flow related to the conditions of the magma based on the crystal differentiation process at the time of eruption has been suggested [11]. Moreover, (iii) the nakhlite could have simultaneously formed with chassignite, in which sulfate- and carbonate-bearing nakhlites are considered to have originated from different lava flows [12]. The mineralogies of nakhlites and olivine-rich chassignites (dunite) markedly distinguish these rocks from other Martian meteorites [12]. Collectively, nakhlite and chassignite Martian meteorites have similar ages of crystallization (1340 ± 40 Ma) and ejection (11 ± 1.5 Ma). Therefore, they are the only Martian meteorites that can be unequivocally identified as coherent igneous rocks from a single source on Mars despite their different mineralogies.

In addition, several models have been offered for investigating the depth profile of the nakhlite parent body and for each nakhlite. For example, three distribution models have been proposed based on (1) the degree of crystal growth of the minerals in one huge lava deposit according to the olivine zoning, the Fe-rich rims shown in the pyroxene, and the amount of the feldspathic mesostasis in the compositional mineral ratios of the meteorite [10,13,14]; (2) the age of iddingsite formation estimated from noble gases trapped and preserved in apatite in mesostasis, which reflects the timing of each lava flow event [11]; and (3) a comparison between the secondary minerals (sulfates and carbonates) produced in each meteorite and the Martian conditions. For example, sulfate would be present near Martian surface nakhlites affected by an oxidizing atmosphere such as that preserved in Miller Range (MIL) 03346 and Yamato (Y) 00 nakhlites, whereas carbonate would have formed in a reductive subsurface region of the nakhlite such as that preserved in Nakhla, Governador Valadares (GV), and Lafayette [15–17]. Distribution model (3) is initially based on model (1) in terms of the primary mineral compositions and distributions. Regarding the origin of nakhlite on Mars, Cohen et al. [11] suggested that a crater present in the Elysium region is a potential source crater for the nakhlites. This crater, located at 130.799° E, 29.674° N, contains preserved ejecta rays [8] indicative of a recent impact event, which is subsequently consistent with the cosmogenic exposure age of 10.7 ± 0.8 Ma obtained for the nakhlites [11]. In addition, the diameter of this crater is 6.5 km, indicating that the impact had sufficient energy for excavating and ejecting materials beyond the orbit of Mars [18]. The numerous sub-horizontal deposition layers present are interpreted as lava flows [11]. Iddingsite texture forms in nakhlites through the aqueous alteration of olivine, which is the focus of this study. Therefore, iddingsite texture is attributed to an aqueous-origin alteration track formed on Mars because (i) the iddingsite structure was cut by fusion crust during its entry into Earth's atmosphere and (ii) the hydrogen isotope value (δD) of iddingsite for a fraction released at high temperature during the step-wise heating of the nakhlites reached 900 permil relative to the Martian surface water having a δD of about 3000–6000 permil [19–21]. The formation age of the iddingsite is 633 ± 23 Ma, at which time surficial water was completely lost on Mars, whereas the age of the bulk nakhlite is 1327 ± 39 Ma as Martian igneous rock [5]. Therefore, it is reasonable to infer that iddingsite formation related to thermal fluid alteration was produced by meteorite impact or unknown processes in the subsurface icy

reservoir [22]. Thus, previous studies have provided evidence indicating that the iddingsite originated from water–rock interaction on Mars (e.g., [16,23,24]). Details of the aqueous environment (e.g., Eh–pH condition) related to iddingsite formation can be found in Bridges and Schwenzer [25], who also reported that the precipitation conditions of Nakhla carbonate (Ca-Fe-Mg siderite) are consistent with 75–100 °C, pH = 4, and water-rock ratio (W/R) = 100 with saponite and serpentine-like phases occurring as partial replacement at low temperatures. Moreover, previous studies reported that the sulfate is a late stage of secondary mineral production in this brine formed by evaporation from the final fluid [25,26].

Herein, we focus on two secondary mineral types found in iddingsite: sulfates and carbonates. Terrestrial calcite, which formed within the Martian vein after its fall on Earth, has been confirmed in some nakhlites such as Northwest Africa (NWA) 998, NWA 5790, NWA 817, and Yamato (Y) 000749 [23]. Previous studies suggest that some sulfate minerals also formed on Earth through terrestrial weathering (e.g., [27,28]). Pre-terrestrial gypsum and jarosite might have also been preserved in the nakhlite meteorites although not in abundance, as evidenced by the low gypsum content in the Nakhla meteorite. The presence of such gypsum and halite in Nakhla indicates that saline solutions were involved in its formation [27]. Once the meteorites were present in Antarctica, terrestrial liquid water could have produced gypsum and jarosite veins along the exposed surface of the stones associated with areas of Antarctic rock varnish (e.g., [15,28]). Such veins of terrestrial sulfates have been observed in some nakhlites such as Y 000749 and MIL 03346 [15,28]. These terrestrial sulfates would have overprinted any pre-existing Martian sulfate such as gypsum or jarosite in the iddingsite to be redissolved by terrestrial water. Such materials are indistinguishable without isotopic analysis. The distinct sulfate vein near the fusion crust is likely of terrestrial origin as refilling in the same process as that of calcite [24]. Pre-terrestrial gypsum and jarosite could have survived in the central portions of the meteorites to be unaffected by terrestrial weathering [16,28]. However, carbonate minerals are reported to have formed on the Martian subsurface/surface and are accompanied by various secondary minerals (e.g., [29]). Siderite veins, rather than calcite, are considered to be of Martian origin and clearly cross-cut the olivine even in areas within the rock that do not contain terrestrial jarosite [17]. The formation processes of these minerals are unclear because the Fe-bearing carbonate should have precipitated under reductive and alkaline conditions, whereas Fe-bearing sulfates, particularly jarosite, precipitate under oxic and acidic conditions. Notably, these two mineral types cannot form simultaneously under equilibrium conditions. Therefore, the simplest explanation for the juxtaposition of such jarosite and siderite is simply that the former is terrestrial in general. The well-weathered Y 000749 shows clear vein features indicating the terrestrial origin of jarosite (e.g., [13]).

Therefore, the aim of this study is to elucidate the formation processes of the two secondary mineral types preserved in the nakhlite through detailed analysis of trace elements in the carbonate. For such examination, we employ micro-focused X-ray fluorescence-X-ray absorption fine structure (μ-XRF-XAFS) analysis using synchrotron-based X-ray microscopy. We analyze the carbonate iddingsite in Y 000593, an Antarctic nakhlite, in which both carbonates and sulfates have been reported in previous studies [16,21]. The analysis of this meteorite via X-ray microscopy enables us to elucidate the detailed alteration environment of nakhlite host rock on Mars by determining the distribution of the trace elements and chemical species.

## 2. Materials and Methods

### 2.1. Meteorite Specimen

For this study, we used a polished thin section (PTS) of Y 000593 (58-1 and 63-3) prepared by the National Institute of Polar Research, Japan (NIPR, proposal no. 1725; Figure 1). This meteorite was collected in the year 2000 in Antarctica near the Yamato Mountain

range and was paired with Y 000749 and Y 000802 according to the textures and noble gas analysis; accordingly, these three meteorites are known as Yamato 00 nakhlites [30]. The terrestrial age of Y 000593 is 11.8 ± 1.0 Ma. Y 000749 was subjected to heavy terrestrial weathering [24]. Different from weathered meteorites found in hot deserts (e.g., Nakhla) [31] and Antarctica (e.g., Y 000749), Y 000593 is suitable for this study. In addition, this meteorite contains both sulfate- and carbonate-bearing iddingsites in the vicinity of olivines [16,21] (Figure 2). The size of the iddingsite in Y 000593 is several to tens of micrometers in width by several tens to hundreds of micrometers in length. Its main components are Fe- and Mg-rich phyllosilicates (smectite, considered to be ferric-saponite); amorphous silicates of saponitic composition (typically treated as gels); and accessory minerals including siderite, Fe oxy(hydro)oxides such as goethite; laihunite; jarosite; gypsum ($CaSO_4 \cdot 2H_2O$); and other carbonates such as Mn-, Ca-, and Mg-bearing siderite (e.g., [15,16,21,23,24]). Amorphous $SiO_2$ (opal) was also reported as a very rare secondary phase [32].

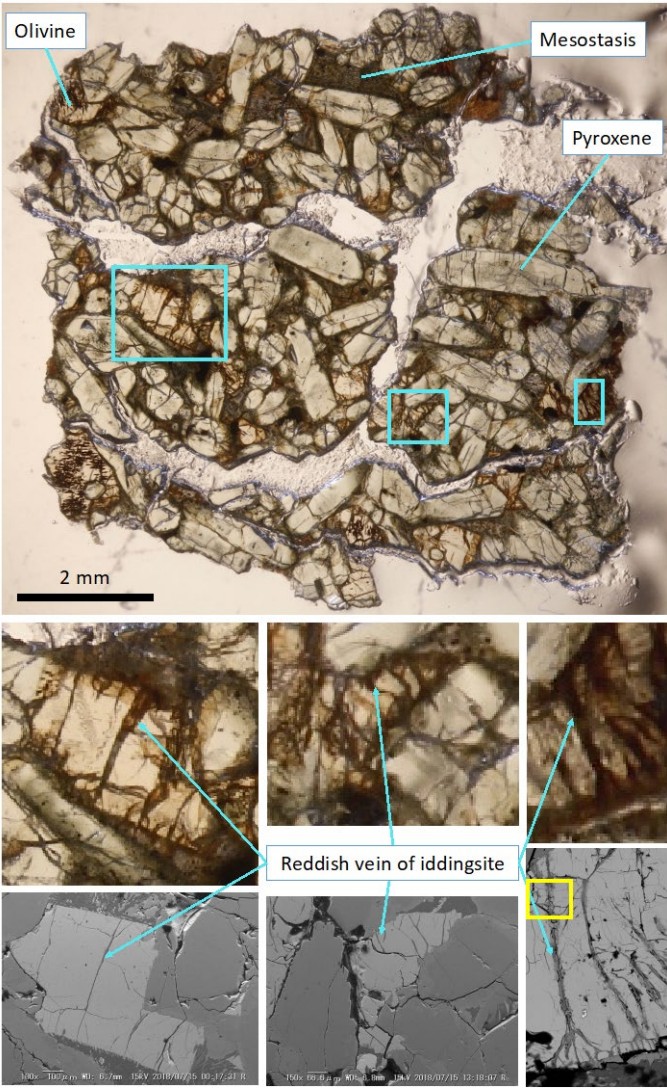

**Figure 1.** Optical microscopy reflected light image of a polished thin section (PTS) of Yamato (Y) 000593 #58-1 analyzed in this study. The light-green columnar mineral is pyroxene; the reddish-brown mineral is iddingsite, which crosses the olivine; and the grayish-black area between them represents feldspathic mesostasis. The light blue square indicates the locations of the main olivine grains analyzed herein; optical microscopy and backscattered electron (BSE) images of the light blue squares are also shown. The yellow square region indicates the area of Figure 2a.

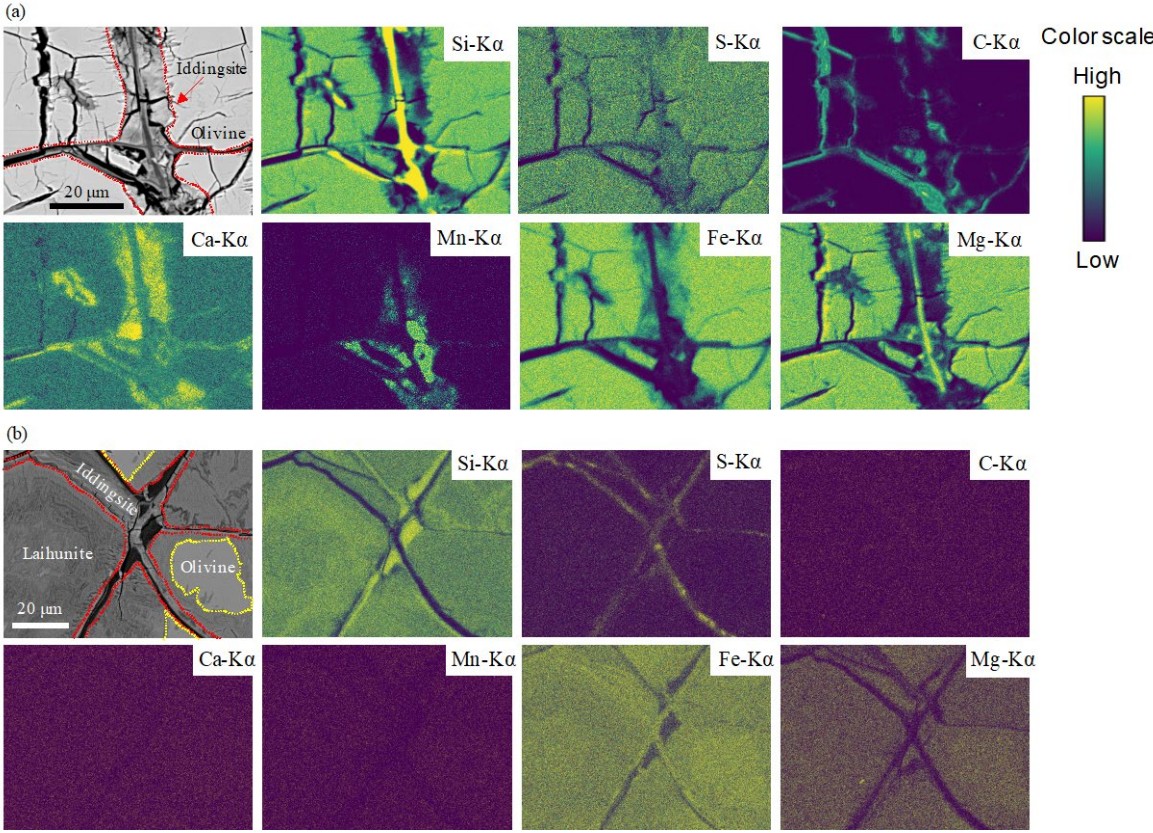

**Figure 2.** Backscattered electron (upper left) and elemental distribution images of iddingsite obtained via field-emission scanning electron microscope-energy-dispersive spectroscopy (FE-SEM-EDS). Iddingsite occurs inside the red line in the BSE image. (**a**) Representative images of carbonate-bearing iddingsite in Y 000593. The central part of the iddingsite is rich in Mg and Si, suggesting the presence of clay minerals. S could not be detected through SEM-EDS because its low concentration was below the detection limit. C was present throughout the iddingsite; the localized area of very high intensity is an effect of the resin used in the preparation of the thin section. Ca was also almost uniformly present throughout the iddingsite, and Mn was concentrated locally. Fe was present throughout the iddingsite, although its intensity was relatively weak owing to the effect of Fe in the olivine. (**b**) Representative images of sulfate-bearing iddingsite in Y 000593. The area within the yellow dotted line is pure olivine, and the area between the yellow and red dotted lines is laihunite. In the BSE image, the Si-rich vein appears to be darker than that in (**a**) because the contrast was adjusted to enhance the laihunite visibility. These images were obtained from the polished tip sample (Y 000593, 120).

### 2.2. Optical Microscopy

A Leica M205C microscope (Leica Microsystems GmbH, Wetzlar, Germany) was used for bright field observation of the Y 000593 PTS, which revealed altered olivine. The reddish vein shown in Figure 1 is indicated as iddingsite.

### 2.3. SEM and EPMA

Detailed identification of iddingsite was performed via field-emission scanning electron microscope (FE-SEM) and field-emission electron probe microanalyzer (FE-EPMA) to confirm the distributions of olivine, pyroxene, and feldspathic mesostasis after the 5-nm-thick C deposition in the PTS. This analysis enabled us to clearly identify the iddingsite. The microstructure of the samples was observed via FE-SEM (JEOL JSM-7000F) and FE-EPMA (JEOL JXA-8530F) using a backscattering electron detector at the University of Tokyo (UT). FE-SEM (JEOL JSM-7100F) was also performed at NIPR (JEOL Ltd., Tokyo, Japan). For both FE-SEM and FE-EPMA, backscattered electron (BSE) images were measured at an acceleration voltage of 15 kV and a current of 50 nA. Point analysis was performed at an acceleration voltage of 12 kV with a current of 6 nA in FE-EPMA. Quantitative analysis of the carbonate iddingsite was conducted through FE-EPMA at UT.

*2.4. Focused Ion Beam*

Ultrathin films of FeS (pyrite or pyrrhotite) were prepared from thin-film meteorite samples for scanning transmission X-ray microscopy (STXM) analysis using a focused ion beam (FIB) system. The equipment employed was an SMI3200 (Seiko Instruments Inc., Chiba, Japan) at the Photon Factory (PF) of the High-Energy Accelerator Research Organization (KEK; Tsukuba, Japan). The sample was lifted using a Mo needle controlled by a manipulation system manufactured by Kleindiek Nanotecnics. A Play Station 2 (Sony Computer Entertainment, CA, USA) controller was used for improving the operability. The location of the ultrathin-film cutting process was determined on the basis of the FE-SEM-EDS analysis results. The Ga beam conditions employed during the processing were as follows: Ufine 80–97 pA, Fine 600–800 pA (originally 300), Mid 700 mA, Rough 3000 mA, and Urough 6700 mA (originally 6500). Finally, a thin, 300-nm thick film was prepared. The Ga beam conditions at the time of measurement were as follows: Acc = 30.0 (kV), Emi = 2.00 ($\mu$A), Ext = about 6.74 (kV), and Sup = about 781–1004 (V). The vacuums of the gas and main chambers were $4.5 \times 10^{-6}$ and $2.9 \times 10^{-5}$ Pa, respectively. W-deposition ($W(CO)_6$) was used for sample protection and for adhering to the pillar of the Mo-transmission electron microscopy (TEM) half-grid (OmniProbe). Ultimately, an ultrathin section of FeS at $15 \times 10 \times 0.1$ $\mu$m in width, height, and thickness, respectively, was obtained to include a mesostasis region.

*2.5. Adsorption Experiment for Cr Adsorption on Mn Oxides in Water*

The adsorption of Cr on Mn oxides, regarded as an important process in this study, is affected by the Mn mineral species, Cr speciation, and pH of the solution. For example, although $Cr^{3+}$ is not adsorbed onto $MnO_2$ owing to the oxidation of $Cr^{3+}$ to chromate [33], it can be adsorbed onto $Mn_2O_3$ and MnOOH [34]. Herein, the following experiments were conducted to investigate the pH dependence of $Cr^{3+}$ adsorption onto MnOOH. First, a $Cr^{3+}$ solution with a Cr concentration of 103 mg/L was prepared from $Cr(NO_3)_3 \cdot 9H_2O$. From this solution, seven solutions were prepared by adding 10 mg of powdered MnOOH (Nichika Inc., Kyoto, Japan). A final solution containing 100 $\mu$g/L Cr (ionic strength: 0.020 M by $NaNO_3$) was prepared by adjusting the pH to 2, 3, 4, 5, 6, 7, and 8 and was stirred at room temperature for 24 h until equilibrium was attained. Preliminary experiments suggested that 8 h were sufficient for attaining equilibrium. The precipitates and supernatant were separated by filtration using hydrophilic polytetrafluoroethylene (PTFE) membrane filters with 0.45 $\mu$m pore size. The oxidation state of Cr adsorbed onto MnOOH was analyzed via XAFS analysis at the Cr K-edge at BL-12C in KEK-PF. The results were compared with micro-X-ray absorption near edge structure ($\mu$-XANES) spectra for Cr in the iddingsite obtained via $\mu$-XRF-XAFS. The initial and filtered (supernatant) water samples were diluted to an appropriate concentration, and the concentration of Cr in the solution was measured through inductively coupled plasma mass spectrometry (ICP-MS) to investigate the pH dependence of the $Cr^{3+}$ adsorption behavior onto MnOOH.

*2.6. Synchrotron Radiation X-ray Analysis*

After observation via FE-SEM-EDS and FE-EPMA, the trace element amounts were measured using synchrotron radiation-based X-ray spectroscopy analyses, which were performed with various beamlines at three facilities: BL-4A, BL-12C, BL-15A, and BL-19A at KEK-PF (Tsukuba, Ibaraki Prefecture); BL37XU at SPring-8 (Sayo-gun, Hyogo Prefecture); and BL4U at UVSOR (Higashi-Okazaki, Aichi Prefecture). The characteristics of each beamline are listed in Supplementary Table S1.

For the $\mu$-XRF-XAFS analyses, the sample was fixed on the X–Y stage, and a coarse step scan was used to identify the rough location of the iddingsite via imaging analysis over a wide area. Then, a fine step scan for $\mu$-XRF imaging analysis was performed to identify the secondary minerals in the iddingsite. In particular, the distribution and

speciation of Cr in the iddingsite were measured at SPring-8 BL37XU, whereas μ-XRF imaging analysis was performed with a step width of 0.1 μm per pixel, an image area of 400 × 400 pixels, and an integration time of 0.2 s/line using an on-the-fly μ-XRF analysis system. The measurement energy was set according to the elements to be measured. We analyzed numerous Mn-rich portions of carbonates in the analysis at BL-4A at KEK-PF to obtain the μ-XRF-XANES results at the Mn K-edge. However, the XANES could not be obtained at the Cr K-edge owing to the limited energy range in the beamline. Thus, we employed BL37XU with a high-order X-ray cut mirror system at SPring-8 to measure the K-edge XANES for Cr in all Mn-rich portions of the carbonates.

STXM was also employed to conduct chemical species imaging by detecting the transmitted X-ray intensity at each point with a step width of 30–70 nm per pixel, image area of 300 × 300–130 × 130 pixels, and an integration time of 1–5 ms/point [35]. STXM is capable of analyzing the chemical species distribution using a smaller beam size with higher resolution than that with μ-XRF-XAFS [36]. The analytical data were processed using aXis2000 software (Version: 22-Jul-2013) to obtain XANES spectra at any point of interest within the analyzed area. The aXis2000 analysis package, written in Interactive Data Language (IDL), is available at http://unicorn.mcmaster.ca/aXis2000.html (accessed on 12 May 2021).

The bulk-XAFS analysis for the Cr adsorbed on Mn oxides was performed at BL-12C (KEK-PF) in fluorescence and transmission modes for the adsorption and standard samples, respectively.

The XANES energy acquisition range was 50–100 eV before and after each absorption edge. The REX2000 analysis package (Rigaku Co., Ltd., Tokyo, Japan) and Athena software were used to treat all XANES spectra data measured in this study.

## 3. Results

### 3.1. Identification of the Carbonate Phase in Iddingsite

The Y 000593 used herein mainly contains clinopyroxene, olivine, and feldspathic mesostasis (Figure 1) with mineral compositions similar to those of other nakhlites [12,16,37]. Iddingsite can be readily found in most olivine grains. The SEM-EDS analysis results indicated that the iddingsite contains carbonate and clay minerals located mainly in the boundary between olivine and iddingsite and in the center of the iddingsite, respectively (Figure 2a). These occurrences are similar to the observation results of clay minerals (ferric-saponite) and poorly crystalline silicates sandwiched by siderite in the iddingsite [24]. Although laihunite coexisting with sulfate has been confirmed, as shown in Figure 2b (e.g., [16]), that with carbonate has not been found, as indicated in Figure 2a (e.g., [21,24]). We investigated the chemical compositions of the iddingsite via FE-EPMA in the plotted carbonate portions shown in Supplementary Figure S1; the carbonate compositions are listed in Supplementary Table S2. Elemental images and triangle elemental plots showed that the carbonates contain Mn-bearing and low- or non-bearing siderite (Figure 3). The Mn-bearing type is heterogeneously distributed within the carbonate in the form of ~5 × 10-μm spots in the iddingsite. The Mn speciation of carbonates was also conducted via Mn K-edge XANES (Supplementary Figure S2). Based on these results, the carbonates were characterized as siderite and Mn-rich siderite.

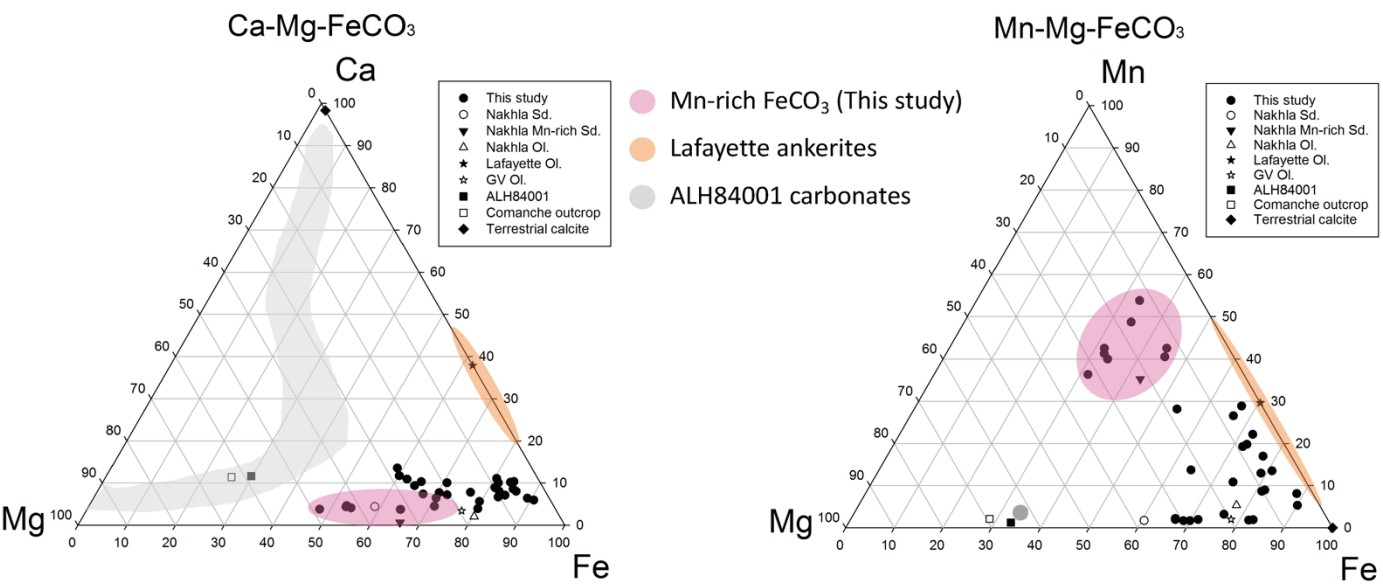

**Figure 3.** Identification of siderite and Mn-rich siderite in the carbonate based on Table S2. Triangular diagram plotting the cationic composition of carbonates using the results of Ca, Mg, Mn, and Fe concentrations in iddingsite quantified by FE-EPMA (analysis spots are shown in Figure S1). The number of analyzed points is 31.

### 3.2. Speciation and Distribution of S in Iddingsite

We detected trace S in the carbonate region, which has been confirmed in previous studies [24], as shown in Supplementary Table S2. The EPMA mapping analysis revealed that S is widely distributed within the carbonate in the iddingsite (Figure 4a). We detected sulfate as the S species in the carbonate-bearing iddingsite by μ-XRF-XAFS analysis (Figure 4b). Pingtore et al. [38] reported that if the sulfate content of the materials is below 1%, determining whether the sulfate is present in the form of micro-inclusions uniformly distributed throughout the sample or as a bulk contaminant at fewer discrete locations is difficult. The post-edge feature of the S K-edge XANES spectra for the S incorporated in the carbonate (Figure 4c) are similar to those of the sulfate in $CaCO_3$, which are present as carbonate-associated sulfate (CAS) [38]. Based on the discussion in Pingtore et al. [38], this similarity reveals that sulfate ions substituted the carbonate site in the carbonate vein. The S-XANES spectra of the CAS were confirmed in the case of more than 3000 ppm sulfate in the carbonate [38], which suggests that our iddingsite vein contains at least 3000 ppm (0.3%) S. Subsequently, the $SO_3$ distribution was determined by EPMA, which showed a maximum concentration of about 0.3 wt.% that decreased from the rim to the center of the olivine in the iddingsite (Figure 4c and Supplementary Table S3). The concentration of S detected here is roughly consistent with that in the clay minerals and gel of iddingsite in other nakhlites [24]. The most likely origin of these sulfates is oxidation of the FeS initially contained in the mesostasis phase because sulfide is the main S species in Martian volcanic rock [39,40].

To confirm the condition of FeS (pyrrhotite) in mesostasis, we investigated the oxidation state of S at FeS in the mesostasis by SEM-EDS, EPMA, and μ-XRF–XAFS (Figure 5a–c). EDS mapping showed that greenish area in the pink region in Figure 5 (FeS) is composed mainly of Fe and $SO_4$. Similar occurrences of FeS have been reported in Treiman [41]. Most of the oxidized FeS have a similar grain appearance, as shown in Figure 5a. We investigated the FeS grains in PTS 58-1 and 63-3, as shown in Supplementary Figure S3, by SEM-EDS. The representative EDS spectra of these oxidized FeS grains are composed of pyrrhotite and oxidized portions, as shown in Figure 5b. The S K-edge XANES spectra obtained from the portion marked in Figure S3 indicates that all measured pyrrhotite had been oxidized as sulfate. Then, we created a 10 μm × 15 μm × 0.1 μm ultrathin section of the mesostasis region of FeS by FIB for the STXM analysis. The

distribution of $SO_4^{2-}$ in the area that connects FeS in mesostasis to olivine was investigated at a finer scale by FIB-STXM. STXM Fe and S L-edge speciation mapping revealed that the sulfate dissolved from FeS migrated within the mesostasis, as shown in Figure 5d,e.

In addition, to confirm the pathway of Fe and S extending from the oxidized FeS in the mesostasis part, elemental imaging was performed near the FeS by EPMA. Figure 6a shows a BSE image of a representative area in which mesostasis and olivine including partially oxidized FeS are in the same field of view. Veinlet structures around the FeS are clearly observed (Figure 6b). For the field of view shown in Figure 6c, the upper limits of the Fe and S images used for the RGB image were changed to optimize the visibility of the low-count part (Figure 6d,e), which confirmed that the distribution of S overlapped with the Fe veinlet part, shown as the white linear structure in the BSE image. Furthermore, the BSE image was magnified in this area to show that this white linear structure is connected to the FeS. Therefore, the fine veinlet structure containing Fe is identical to that containing $Fe^{3+}$ (Figure 6g) and $SO_4^{2-}$ leached from FeS (as shown in Figure 5d,e) and is considered to be a channel that carried the $SO_4^{2-}$ leached from the FeS to the olivine. Although precise quantitative analysis by EPMA could not be performed owing to the small size of the structure, it is suggested that the Fe mineral is Fe (hydro)oxide (e.g., goethite or akaganeite) because it mainly comprises Fe, O, and trace Cl (green part in Figure 6f). Unfortunately, we were unable to detect areas in which the channel was directly connected to the iddingsite in the thin section. However, because the channel and the iddingsite are unevenly distributed in the meteorite and are distributed in the three-dimensional (3D) structure, they could be connected in areas other than our observed surface. Based on this observation, we conclude that the trace amount of S observed in the iddingsite originated from sulfate ions generated by the oxidation of FeS in mesostasis. Previous research also supports that the Fe-enriched veinlets crossing the pyroxene imply that the Fe was mobile in the subsolidus, possibly in Fe-S fluid [42].

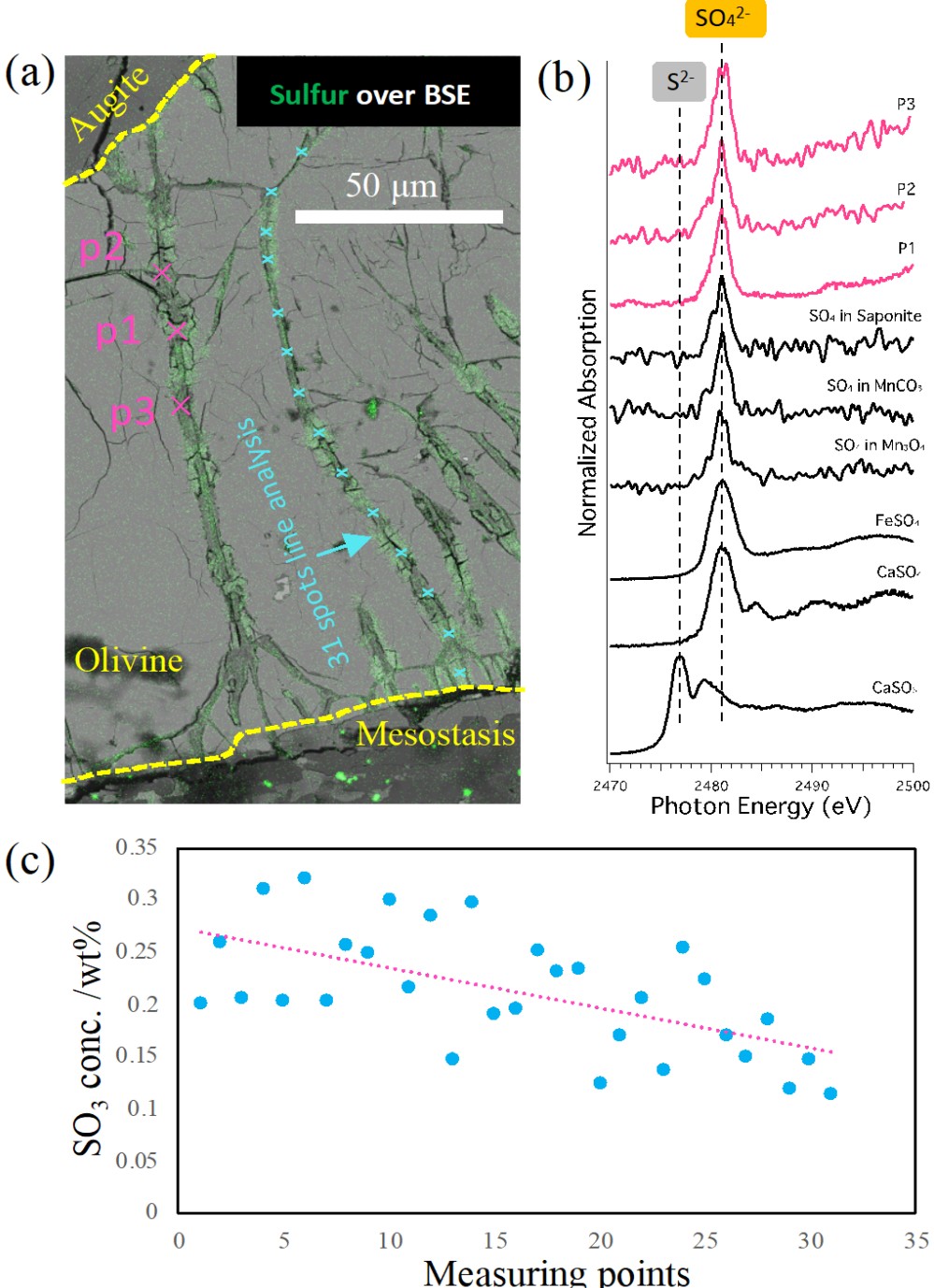

**Figure 4.** Analysis results of S mapping. (**a**) S elemental image obtained via EPMA (green) over the BSE image. The S is universally distributed in the carbonate iddingsite. (**b**) S K-edge XANES spectra obtained from the spots shown by pink cross marks in (**a**) p1–3. The spectra of standard materials are also shown. (**c**) SO$_3$ concentration by EPMA obtained from the light blue cross-marked region in (**a**).

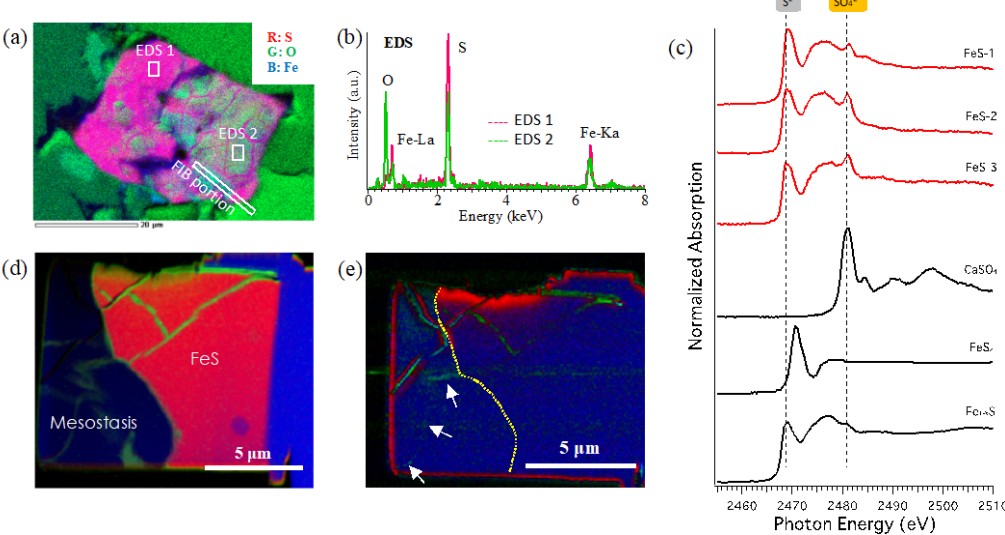

**Figure 5.** Results of S in mesostasis analysis conducted to determine the origin of S in the carbonate iddingsite. (**a**) RGB composite elemental image obtained via SEM-EDS. The selected FeS in the mesostasis next to iddingsite-bearing olivine is shown. The red, green, and blue indicate S, O, and Fe, respectively; the pink area indicates FeS; and the green area surrounding the FeS corresponds to feldspar. (**b**) SEM-EDS spectra obtained from the white square area in the pink region in (**a**) and that in the greenish portion in the pink area of (**a**). The greenish portion mainly comprises Fe, O, and S. (**c**) S K-edge XANES spectra obtained through μ-XRF-XAFS from the FeS in the mesostasis including grains such as those in (**a**). The analysis points are shown in Figure S3. The analysis beam size was about 15 μm. Although detecting the $SO_4^{2-}$ separated only from $S^{2-}$ is difficult, the $SO_4^{2-}$ peak is clearly confirmed. (**d**) RGB composited image of Fe oxidation states by STXM. Red, green, and blue colors indicate FeS, $FeSO_4$, and mesostasis, respectively. The thin section was excavated from the white square region in (**a**). (**e**) RGB composite image of S oxidation states by STXM. The thin section was excavated from the white square region in (**a**). Red, green, and blue colors indicate sulfide, sulfate, and weak X-ray transmitted area including sulfide, respectively. Here, the yellow dotted line is the boundary between FeS and mesostasis, where the FIB section is too thick to measure the transmitted X-ray at the S L-edge energy. $S^{2-}$ was detected only in the upper part of the FIB section with smaller thickness. The white arrows indicate the flow pathway of sulfate to mesostasis.

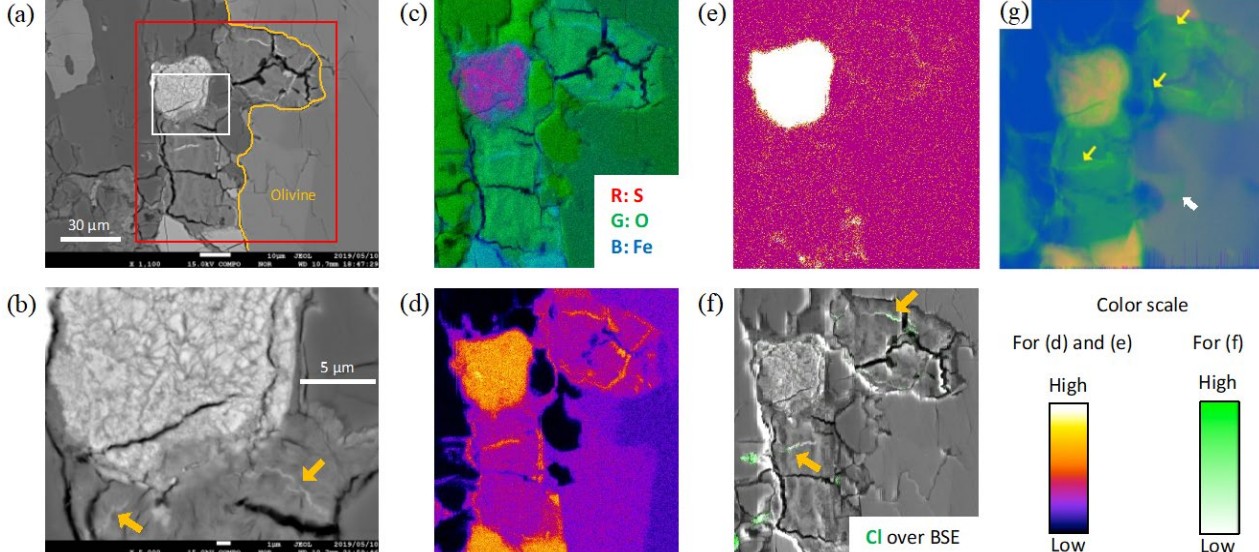

**Figure 6.** Survey of $SO_4^{2-}$ flow in the mesostasis and olivine areas by EPMA. (**a**) BSE image of a typical area showing mesostasis and olivine containing oxidized FeS in the same field of view. White square indicates the field of view in (**b**). Red square indicates the field of view in (**c**) to (**g**). (**b**) Magnified BSE image around FeS in (**a**). The orange arrows indicate the vein-like structures that migrated from FeS. (**c**) RGB image of the area near the FeS in the field of view in (**a**). Red,

green, and blue indicate S, O, and Fe, respectively, and the purple color indicates FeS. (**d**) Fe distribution in the same field of view as that in (**c**). (**e**) Distribution of S (white area) in the same field of view as that in (**c**). The color balance was adjusted to enhance the low-count area. (**f**) Cl distribution (green) superimposed on the BSE image. Although its intensity is low, the Cl overlaps with the vein found in (**d**) (e.g., orange arrows). (**g**) Fe valence distribution image. Red, green, and blue correspond to $Fe^{2+}$, $Fe^{3+}$, and Fe-free areas, respectively. The yellow arrows indicate part of the micro vein that might be the pathway for the dissolved S. The mapping energies of $Fe^{2+}$ and $Fe^{3+}$ are 7117 and 7131 eV, respectively, in the Fe K-edge XANES. The white arrows indicate the flow of this fluid into the olivine, although this was not clearly shown on the surface of the PTS owing to the 3D structure of the nakhlite.

### 3.3. Characterization of Cr in the Mn-Bearing Siderite

We detected Cr in the Mn-rich siderite, as shown in Figure 7. The μ-XRF-XAFS mapping of Mn carbonate revealed trace Cr in almost all of the Mn carbonates, thus indicating a correlation between the Cr and Mn distributions at the micrometer scale. Cr has also been detected in Mn-rich siderite in Nakhla in previous research, as shown in Supplementary Table S2 [26]. It is widely accepted that Cr is concentrated in pyroxene during the crystallization process of basaltic magma in a mafic-ultramafic system [43]. Because the fluid related to iddingsite formation can flow through pyroxene, dissolution of pyroxene is a likely origin for the Cr and Ca; in particular, up to 9000 ppm of Cr can be contained in the pyroxene in terrestrial basaltic rock [43]. Moreover, Cr is highly compatible with pyroxene, and its distribution (zoning) and zoning type in each pyroxene crystal are strongly affected by the formation environment of the pyroxene [43]. This zoning was noted in the pyroxene of our sample between the Cr-rich cores and Cr-poor rims (Figure 7c), reported as "sector zoning" in Schoneveld et al. [43]. This sector zoning has been reported in previous studies of Y 000593 [44,45]. The decrease in Cr is correlated with an increase in Ti from the core to the rim, as described in Schoneveld et al. [43] and Mckay et al. [44]. According to a rough estimate of Cr concentration of about 4500–6000 ppm in the Cr-rich part [43], its concentration in the iddingsite is between 1600 and 8700 ppm.

The Cr K-edge XANES results indicate that the Cr in the Mn carbonate has a main peak corresponding to Cr(III). Moreover, the spectra did not show pre-edge peaks that specifically reflect the presence of Cr(VI) at 5988 eV ($1s \rightarrow 3d$–$4p$ hybrid orbital transition [46]). These results clearly show that the Cr in the $(Mn,Fe)CO_3$ is trivalent. Cr(III) is also the dominant valence state in pyroxene, according to XANES analysis of Martian basaltic shergottites because Cr(III) is much more compatible than Cr(II) in the pyroxene structure [47]. The XANES spectrum of Cr-bearing enstatite (clinopyroxene) shows a prominent shoulder at 5996 eV that precedes the absorption edge at 6001 eV [48]. The Cr K-edge XANES in the Mn carbonate is similar to that of Cr in pyroxene (blue-colored spectra in Figure 7d), particularly when the Cr(II) component in the pyroxene is subtracted. This could be related to the degree of the spectrum feature around 6001 eV, exhibited as a peak or shoulder.

### 3.4. Behavior and XANES Features of Cr Absorbed Onto MnOOH

We conducted adsorption experiments of Cr(III) onto MnOOH (Figure 8). According to the distribution coefficient, adsorption onto MnOOH was observed mainly at pH > 6, which suggests that the fluid shifted to circumneutral pH during the formation of MnOOH. The Cr K-edge XANES was also measured for Cr adsorbed onto MnOOH, which showed that Cr was adsorbed as Cr(III). This adsorption behavior of Cr indicates that the Mn was initially deposited as Mn(III) oxides (MnOOH), which was subsequently transformed into Mn(II) carbonate during the activity of the carbonate-bearing fluid. According to the XANES spectra in Figure 8b, the coordination environment of the Cr that adsorbed onto MnOOH was similar to that of Cr in the $(Mn,Fe)CO_3$ in the iddingsite. Moreover, the spectra for Cr in the $(Mn,Fe)CO_3$ are clearly similar to those for Cr(III) incorporated in geological carbonate (Ediacaran carbonate) in the work of Fang et al. [49], having one prominent peak at 6007 eV, a weaker peak at 6021 eV, and a shoulder at about

6002 eV. This similarity also supports that Cr(III) could have been present in the precursor Mn mineral that was incorporated into the carbonate, as confirmed by Cr K-edge XANES.

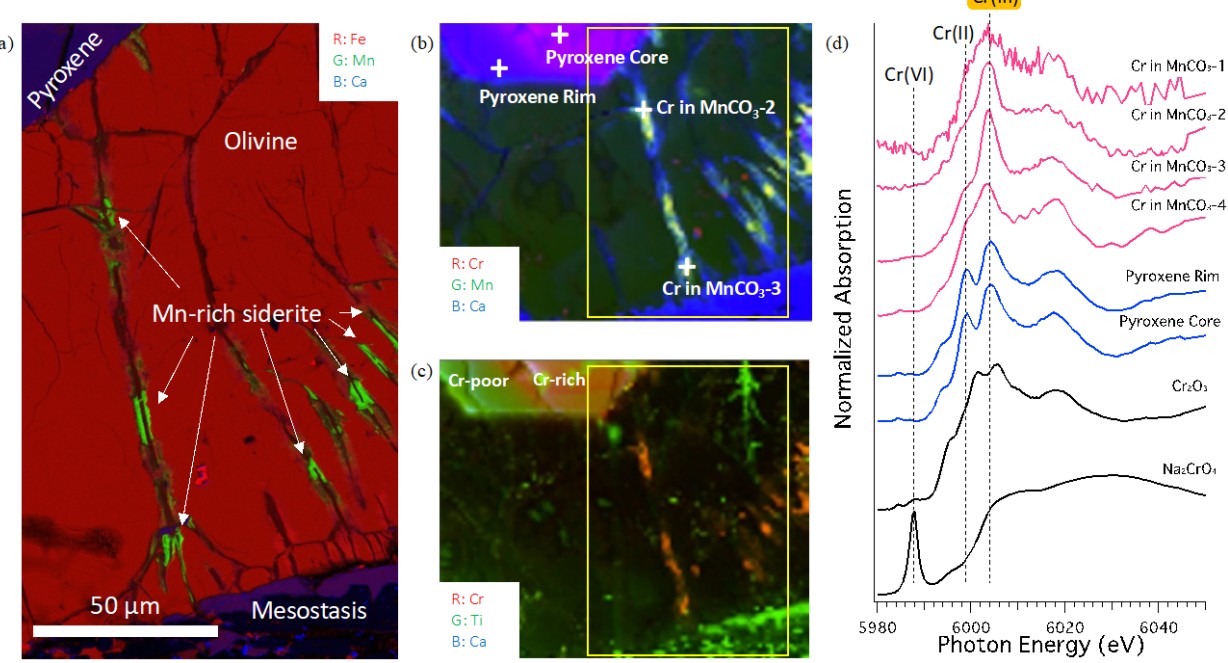

**Figure 7.** μ-XRF-XAFS analysis results focusing on Mn and Cr in Mn-rich siderite. (**a**) RGB composite image of representative olivine containing carbonate iddingsite by EPMA. Red, green, and blue colors indicate Fe, Mn, and Ca, respectively. The heterogeneous greenish area indicates Mn-rich siderite (MnFeCO₃). (**b**) RGB elemental image of the same area shown in (**a**) by μ-XRF-XAFS. Red, green, and blue colors indicate Cr, Mn, and Ca, respectively. The yellow part is Mn-carbonate, which clearly contains Cr in the Mn-bearing carbonate. (**c**) Red, green, and blue colors indicate the distributions of Cr, Ti, and Ca, respectively. Ca is also found in the iddingsite. The yellow square indicates the view of (**a**). (**d**) Representative Cr K-edge XANES spectra obtained from the Mn-rich siderite carbonate in the portion marked by "+" in (**b**) and in Figures S4c and S4e. The XANES spectrum of the yellow part shows peak energy similar to that of $Cr_2O_3$, which suggests that the Cr is Cr(III) in the Mn-rich siderite.

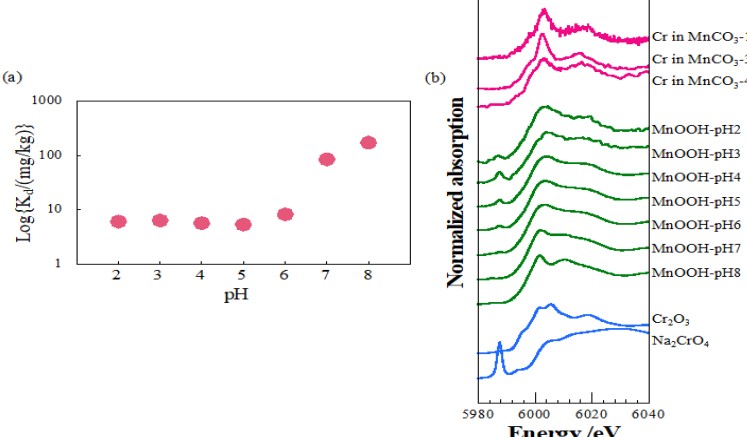

**Figure 8.** Results of Cr–MnOOH adsorption experiment. (**a**) Distribution coefficient of $Cr^{3+}$ to MnOOH plotted as a function of pH. The vertical axis shows the concentration of Cr in the MnOOH (solid phase) relative to the concentration of Cr in the liquid phase in the logarithmic scale. At pH < 6, the Cr was not incorporated into the solid phase. The adsorption began at pH = 6 and increased at pH > 7. (**b**) Cr K-edge XANES spectra obtained from solid phase samples at each pH level. For comparison, the Cr-XANES spectrum of the Mn-enriched area in the iddingsite is shown in pink. The spectra of trivalent and hexavalent Cr compounds are shown in blue as reference samples.

## 4. Discussion

### 4.1. Characterization of Siderite and Mn-Rich Siderite in Y 000593

A definitive mineralogical composition of the carbonate in Y 000593 has not been previously obtained. Only a minor abundance of [13]C-rich carbonate has been reported in relation to the past Martian atmosphere in Y 000593 [50]. In the present study, we found carbonate in the iddingsite of Y 000593 by SEM-EDS, and EPMA was used to characterize it as siderite and unevenly distributed Mn-rich siderite. The composition of siderite and Mn-rich siderite is similar to that of carbonate in Nakhla olivine fractures (iddingsite) reported previously [26,51–53]. Although the carbonate in Lafayette also contains a high Mn content similar to that of ankerite (orange region in Figure 3), the composition of Mn-rich siderite herein follows a trend similar to that of Mn-rich siderite in Nakhla olivine (pink region including Nakhla Mn-rich Sd., marked as ▼ in Figure 3) rather than Layafette ankerite. Both compositions of siderite and Mn-rich siderite in this study are metastable, which implies that the alteration event of its formation fluid occurred at low temperature and that the $HCO_3^-$ in the fluid was rapidly depleted. In the model of nakhlite accumulation formation, Nakhla and Y 000593 are considered to exist at neighboring depths, with the former occurring deeper than the latter [10]. The siderite and Mn-rich compositions are highly consistent with those of Nakhla, which suggests that the carbonates in the iddingsite of Y 000593 were formed through alteration by $CO_2$-rich fluids. It is noteworthy that the Mn-rich siderite found in Nakhla has also been found in Y 000593. If the Mn in the two samples has the same origin, it could have been supplied from the upper part of the rock (from the direction of the Martian surface) because the pure concentration of Mn-rich siderite in Y 000593 is higher than that in Nakhla, as shown in Supplementary Table S2. Bridges and Schwenzer [25] quantitatively determined the fluid chemistry based on the observed mineralogy of a nakhlite meteorite. They determined that the nakhlite parent rocks on Mars encountered a $CO_2$-rich hydrothermal fluid at 150 °C ≦ T ≦ 200 °C with pH = 6–8 and W/R ≦ 300 as initial conditions. In addition, they reported that the formation fluid conditions of Nakhla siderite are consistent with 75–100 °C, pH = 4, and W/R = 100, with saponite and serpentine-like phases indicating partial replacement at lower temperatures. This similarity in carbonate features suggests that the siderite of Y 000593 was formed by the involvement of fluids similar to those of the siderite in Nakhla.

### 4.2. S Chemistry

Although the coexistence of laihunite and sulfate has been confirmed, as shown in Figure 2b, that with carbonate has not been found, as shown in Figure 2a. Such information can be useful for estimating the formation processes of carbonate and sulfate in iddingsite. The three-fold (3M) structure of the laihunite along the walls of cracks of indicates a heating event at temperatures of 400–800 °C [16] because terrestrial laihunite is known to be a high-temperature alteration phase at this temperature range [54]. Laihunite is formed by a rapid oxidation reaction in olivine cracks generated by impact-induced hot gas (acid fog). Previous oxidation experiments using fayalite at 400–700 °C for 5–480 h clarified that it initially transforms to the 3M structure, with further oxidation producing a 2M structure [55]. In addition, the relatively short heating duration of <3 h reported previously could have prevented the oxidation progression of olivine to form the 2M phase [56].

For the sulfate in the iddingsite, some of the clay minerals and gel in the contaminated veins contain high S contents of 0.5–4.5 wt.% equivalent to up to 10.7 wt.% $SO_3$ [24]. In general, the simplest explanation for these sulfates is terrestrial contamination after falling on the Antarctic ice sheet. Such a high concentration up to 10 wt.% S could not be confirmed in our results, and no S concentration gradient was present between the central clay minerals and the surrounding carbonates (Supplementary Tables S2 and S3). Previous studies indicated that the sulfate can evaporate at a late stage of secondary

mineral production of brine, which formed the carbonates in the iddingsite on Mars [25]. In addition, the island-like distribution of Mn-rich siderite cannot be explained by simple terrestrial contamination such as that shown in Figures 2 and 7.

Regarding the origin of these trace sulfate, oxidized pyrrhotite grains are considered because mesostasis is present in the vicinity of almost all olivines in the nakhlite. Regarding the FeS minerals in the nakhlite, pyrite initially forms below 743 °C and is partly replaced by pyrrhotite ($Fe_{1-x}S$) [57]. Typically, the dissolution of sulfide into water produces acidic fluid by oxidation from FeS to sulfate species (e.g., [58]). Thus, sulfide oxidation could have caused the $SO_4^{2-}$ formation and concomitant acid production [40]. Therefore, the presence of sulfate in the carbonate strongly suggests that these sulfates occurred prior to the carbonate formation because carbonate is readily dissolved by acidic fluid, which cannot coexist with an acidic solution containing sulfate under equilibrium conditions.

Herein, the veinlet pathway of the dissolved sulfate was from oxidized pyrrhotite. Nearly all pyrite and pyrrhotite show significant exchange of S isotopes with external reservoirs such as those on Mars [57,59,60]. Moreover, the variations of sulfate $\Delta^{33}S$ among the nakhlites imply differences in the compositions of their alteration fluids related to their depth-dependent locations [39]. These results indicate that FeS in the mesostasis was partly oxidized to sulfate in fluid occurring locally at each depth of the nakhlite body on Mars.

Our analytical results indicate that the sulfate species in the carbonate in the iddingsite within the olivine originated from dissolution of $SO_4$ from FeS and migrated within the iddingsite area.

$$\text{FeS} + \frac{5}{2}\text{H}_2\text{O} + \frac{3}{2}\text{O}_2 + \frac{1}{3}\text{K}^+ \rightarrow \frac{1}{3}\text{KFe}_3(\text{SO}_4)_2(\text{OH})_6 + \text{H}^+ + \frac{1}{3}\text{SO}_4{}^{2-} \tag{1}$$

In Equation (1), FeS is easily oxidized by liquid water and produces sulfate minerals such as jarosite ($KFe_3(SO_4)_2(OH)_6$). In addition, this reaction causes the sulfate-bearing fluid to be more acidic. This result supports partial oxidation (sulfate formation) of FeS in the mesostasis and are consistent with other observational results of jarosite and natrojarosite in iddingsite [16,61]. Moreover, it is implied that the $SO_4$ (or jarosite) created by this reaction should be origin of the $SO_4$ preserved in the carbonate iddingsite. These trace $SO_4$ particles are very fine (<10 nm) and are separately distributed; they are likely not visible in such a high-resolution TEM analysis and are observed only by S-XANES [62]. The origin of these sulfates present in the iddingsite is likely related to the reaction discussed above. We also speculate that the Fe in the iddingsite could have originated from FeS by this reaction or from the dissolution of olivine. If the carbonate was formed prior to the sulfate formation, it had to be dissolved by acidic fluid. In such a case, carbonate would not be found in iddingsite [63]. Therefore, the acidic and oxidative water flowed first, followed by the alkaline reductive fluid flow.

### 4.3. Mn and Cr Chemistries

As shown in Figure 3 and Supplementary Table S2, a heterogeneous distribution of Mn was found in the carbonate of the iddingsite ($FeCO_3$) and in the Mn-enriched area consisting of Mn-bearing $FeCO_3$ with an Mn concentration of about 50 mol%, as given in Table S2. Homogeneous distribution of Mn would be expected within the carbonate of the iddingsite if the Mn was initially deposited as Mn-carbonate from the dissolved $Mn^{2+}$ in the fluid. Therefore, explaining these heterogeneous Mn distributions by a simple single deposition process is difficult. The S analysis discussed above suggests that the alteration process for producing iddingsite occurred initially under oxic and acidic conditions, although the Eh-pH condition could have shifted to reductive and alkaline fluid conditions by further reactions with olivines and other minerals. Considering this Eh-pH pathway, $MnO_2$ and MnOOH are considered to be precursors of the Mn mineral to provide enrichment of Mn within local areas, which finally led to the heterogeneous

distribution as $(Mn,Fe)CO_3$ by transformation of $MnO_2$ and $MnOOH$ to Mn-bearing $FeCO_3$. A previous study reported that Mn-concentrated portions in mesostasis mainly contain hausmannite ($Mn^{2+}Mn^{3+}_2O_4$), trace amounts of manganite ($\gamma$-$MnOOH$), and rhodochrosite ($MnCO_3$) [64], which is consistent with a possible alteration process from Mn oxides to Mn carbonate suggested in the present study.

To support this process, we also confirmed the chemical processes for the formation of iddingsite based on the Mn-Cr relationship shown in Figure 7a–d. The sector zoning of Cr in the pyroxene shown in Figure 7b,c occurred as a result of chemistry differences by the kinetic effect of a crystal at the growth surface in which elements are incorporated into the structure at different rates at different crystallographic surfaces rather than by changes in chemistry or conditions of the surrounding magma [45,65]. Cr-XANES spectra obtained from both Cr-rich and Cr-poor regions shown in Figure 7d show the same oxidation state and spectra features. This result also supports sector zoning. For example, abrupt zoning is caused by changes in the growth environment of the pyroxenes or those in the chemistry of the surrounding magma, where the budget of Cr has been exhausted or the Cr species has changed to a less-favorable oxidation state for intake to pyroxene, resulting in sudden changes to a lower partition coefficient [43]. Therefore, the sector zoning was likely caused by the crystallization of pyroxenes after the magma system separated from a freely convecting magma system to equilibrium crystallization in a restricted closed system such as a lava flow. Such a lava flow could have originated from the eruption of Olympus Mons on Mars, which became the host cumulate rock of the nakhlite.

Regarding the Cr-XANES obtained from Mn-rich siderite, because the Cr in the $(Mn,Fe)CO_3$ is Cr(III), which is incompatible in the carbonate, we can assume that the Cr(III) was not incorporated into carbonate directly from the water that produced carbonate but instead was affected by Cr contained in some precursor minerals. The origin of Cr in Mn-rich siderite is thought to dissolution from pyroxene, as described above. The Cr distribution in the carbonate of the iddingsite is restricted to the $(Mn,Fe)CO_3$ phase. Therefore, we speculate that the Cr was hosted by Mn minerals before the formation of the iddingsite.

$MnOOH$ and $MnO_2$ are general secondary Mn minerals that can be precipitated from acidic and relatively oxic fluids. Therefore, they are candidates for Mn-enriched minerals that formed prior to the carbonate in the iddingsite. Therefore, the Mn minerals can be evaluated based on the adsorption behavior of Cr onto these minerals [66]. It is widely accepted that Cr(III) adsorption and coprecipitation onto $MnO_2$ rarely occur for two reasons: (i) $MnO_2$ is a strong oxidant that oxidizes Cr(III) to $Cr(VI)O_4^{2-}$, which is highly soluble with lower affinity for various mineral surfaces, and (ii) both $MnO_2$ and $CrO_4^{2-}$ have negative charge at the surface, which inhibits adsorption of Cr(VI) onto $MnO_2$ [33]. In contrast, Cr(III) readily adsorbs onto or coprecipitates with $MnOOH$ [34]. Our results, shown in Figures 7 and 8, indicate the presence of these precursors and Cr(III) adsorption onto them. Figure 8 shows that $MnOOH$ as a possible precursor adsorbed Cr(III) near the natural pH level and that Cr-XANES feature of Cr(III)-$MnOOH$ is quite similar to that of Cr-XANES obtained from Mn-rich siderite in the iddingsite of Y 000593. The feature of Cr-XANES shown in the Cr(III)–$MnOOH$ and Mn-rich siderite has a XANES feature of Cr(III) similar to that incorporated in geological carbonate (Ediacaran carbonate) reported by Fang et al. [49].

The Cr and Mn chemistries discussed above suggest that $MnOOH$ and $Mn_2O_3$ generated by the dehydration of $MnOOH$ are considerable precursors of Cr(III)-bearing Mn carbonate, which can be subsequently produced by the reduction reaction of Mn oxides with $HCO_3^-$-rich reductive fluid. Mn-oxides and Mn-hydroxides have been found in the Martian meteorite and in the sediments of the Martian surface sampled by the Rover exploration vehicle [67]. It is generally believed that the exposed fracture surfaces were formed by aqueous alteration veins beneath the Martian surface [68]. Based on the Rover's observations and the trace element behavior in a laboratory experiment, the Mn (hydro)oxides found on the Martian surface are suggested to be $MnO_2$ [66]. If $MnO_2$ at the

Martian surface was dissolved by fluid and reduced to Mn(III) by the properties of the fluid chemistry, this fluid can be the origin of the precursor Mn(III) minerals in the iddingsites. Possibly, Mn minerals such as those found on the surface were physically transported into the interior of the vein by fluid flow. If the Mn oxides that can be considered as the precursor of the Mn carbonate in this study are associated with the surface materials, Mn can be used for further clarification of the Martian surface conditions.

### 4.4. Reaction Pathway During Iddingsite Formation

Based on the discussion presented in the previous section, the results of this study indicate that the early sulfate-forming fluids were strongly involved prior to the carbonate formation in the iddingsite. If the present chemical composition of the iddingsite was formed by reactions in one type of fluid, the acidic fluid had to dissolve the carbonate to form sulfate. No carbonate ion source is present in the nakhlite meteorite that could change the fluid composition on such a large scale. Therefore, it can be reasonably assumed that at least two types of fluid with different chemical compositions are responsible for the formation of the iddingsite found in our specimen. Here, we propose reaction pathways for the formation of the iddingsite, as illustrated in Figure 9.

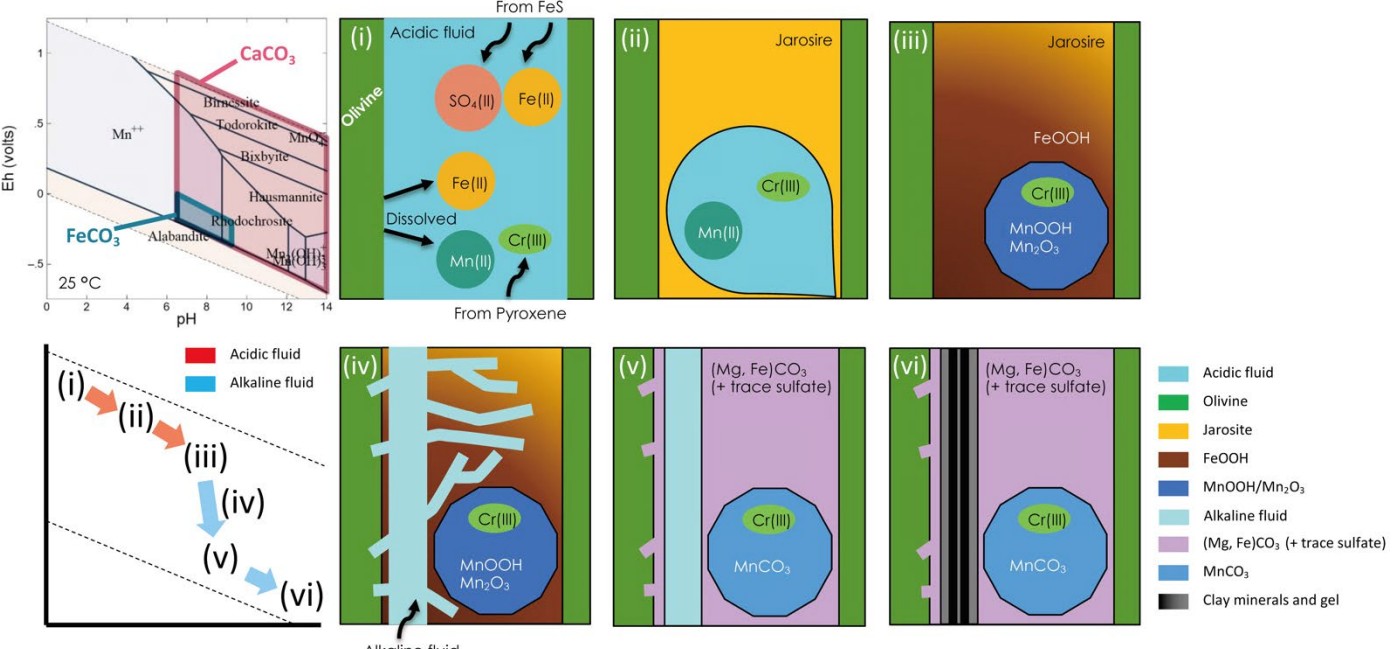

**Figure 9.** Schematic illustrations of the transition process in iddingsite based on Eh-pH.

(i) Sulfide dissolution: First, an acidic solution containing Fe and $SO_4^{2-}$ dissolved from FeS in mesostasis caused the dissolution of the olivine surface within the grain boundary region. Cr and Ca were dissolved from pyroxene by this fluid. Fe was dissolved mainly from the olivine and partly from the pyroxene, and Mn was likely dissolved from olivine and manganese species at the Mars surface to be transported to the vein.

(ii) Sulfate mineral formation: After the formation of the fluid, as discussed in (i), the dissolved $Fe^{3+}$ and $SO_4$ formed jarosite under oxic and acidic condition at almost same time as the process described in (i). Using thermodynamic calculation, we estimated that such fluid formed sulfate minerals such as jarosite at about 200 °C [40], which has been reported as sulfate-type iddingsite. However, it is likely that Mn could not be incorporated into the jarosite but instead remained in the residual water. This was observed during formation of jarosite in a terrestrial environment (e.g., [69]).

(iii) Goethite formation: After the formation of jarosite with complete consumption of the sulfate, $H^+$ was consumed by the subsequent dissolution of olivine, and the pH increased, which produced Fe hydroxide precipitates in place of the jarosite. Below 200 °C, the jarosite continued to transform into FeOOH (hematite or goethite) until equilibrium was established [40]. Consequently, the jarosite became goethite. Under conditions of Eh = 0.5–1.0 (V) and pH = 6–9, Mn can be also precipitated as $Mn_2O_3$ or MnOOH during the transition of the Eh-pH condition [70]. This process is supported by (a) the heterogeneous distribution of Mn-rich areas within the carbonate of the iddingsite and (b) enrichment of Cr(III) by adsorption or coprecipitation to MnOOH.

(iv) Secondary $CO_2$-bearing reductive alkaline fluid generation: This fluid is the source of the carbonates and clay minerals described below and could have been caused by the residual heat of the primary fluid. The water reservoir would have been charged with $CO_2$ from the Noachian atmosphere [17]. Because clay minerals contain Na and P, part of this hydrothermal fluid composition is most likely derived from the dissolution of feldspar and apatite in mesostasis [15]. No carbonate was present in the iddingsites in some nakhlites with only clay minerals/gel penetration, which could serve as evidence of rapid carbonate depletion. This depletion likely occurred because the supply of carbonic acid was limited or was cut off from the fluid source. The subsequent formation of siderite and clay minerals/gel caused by the involvement of this fluid is thought to have been a short-term reaction owing to the metastable compositions of these minerals [24]. Alternatively, this reaction could have been caused by an impact event when the nakhlite was ejected from Mars. Even after the carbonate was depleted, this fluid continued to penetrate the host rock while lowering the W/R and reacting with the surrounding minerals to form clay minerals/gel. Then, this hydrothermal system was terminated by the precipitation of silicate gel and evaporation of salts such as halite [15].

(v) Siderite formation: Nakhlite parent rocks on Mars encountered a $CO_2$-rich hydrothermal fluid at 150 °C $\leqq$ T $\leqq$ 200 °C, pH = 6–8, and W/R $\leqq$ 300, However, the formation fluid condition in Y 000593 needs to be 75–100 °C, pH = 4, and W/R = 100 to form the Nakhla-like siderite found in this study [25]. Although Fe and Mn were present as (hydro)oxide, the $CO_2$-bearing reductive water attacked these precursor minerals, which resulted in their transformation into carbonate iddingsite with ultimately Eh $\leqq$ 0.2 and pH = 9–11. This groundwater was likely rich in carbonate ions [17,21]. By reaction with the carbonate-bearing fluid, FeOOH and MnOOH/$Mn_2O_3$ formed $FeCO_3$ and $MnCO_3$, respectively. It is speculated that the Cr(III) incorporated in the MnOOH was retained in the $MnCO_3$, as observed in the XANES spectra. Sulfate ions remaining in the jarosite were incorporated into the carbonate, as revealed by the µ-XRF-XAFS and FE-EPMA analyses conducted herein.

(vi) Poorly crystalline clay minerals and amorphous silicate gel formation: Finally, clay minerals such as smectite and amorphous silicate gel formed in the center region of the iddingsite. In the case of Nakhla, the fluid precipitated Mg,Fe-rich siderite at 150–200 °C followed by crystalline smectite (saponite) formation at 50 °C, pH 9, and W/R = 6, followed in turn by rapid precipitation of an amorphous silicate gel [25]. This fluid was enriched in the most soluble species (e.g., K, Na), of alkaline pH, and similar to those in terrestrial conditions, indicated that the terrestrial seawater did not contaminate the poorly crystalline clay minerals and amorphous silicate gel [24]. The plagioclase in the mesostasis contains a significant amount of Al, whereas the plagioclase replaced by smectite and amorphous silicate contains less Al [64]. Therefore, smectite and amorphous silicate gel are thought to have been formed by reaction of alkaline fluids (pH < 12) containing dissolved Fe and Mg ions to the plagioclase. Of the clay minerals, saponite and serpentine were identified, which are considered to be in local heterogeneous distribution in relation to those of the surrounding minerals [24]. These detailed variations such as the trace element distribution of clay minerals and gel could be the result of variations in the composition minerals, W/R, and temperature of the surrounding fluid [15,24].

Although some of the proposed processes are speculative, the main concept of the process is the explanation of the carbonate phase with the concentration gradient of sulfate ions incorporated in carbonates from FeS to the olivine grain boundary. If the order of the fluids flowing within the veins were reversed, dissolution of FeS would have promoted the acidification of the iddingsite, leading to complete dissolution of the carbonate [63]. Although a definitive origin of the sulfate- and carbonate-bearing fluids remains unclear, this study clarifies circumstantial evidence indicating the order of the fluids involved in the nakhlite formation.

*4.5. Estimation of the Alteration Process in Nakhlite*

For the driving force of the initial fluid formation, it is reasonable to consider impact-induced ice melting and natural melting of surface ice during a temporary warm period on the Martian surface. Considering the alteration periods of the nakhlite, the most likely cause is impact-induced hydrothermal alteration of the nakhlite pile at the margins of an impact crater [25]. It is unlikely to be a reaction of the remaining water in the past Martian ocean considering the formation age of nakhlite at 1.3 Ga. Combined with the results of previous studies (e.g., [9,15,16,24,25,40,53]), the alteration processes of Y 000593 in our model after the crystallization of nakhlite are used consider the sequences given below and in Figure 10.

1. Impact event: An impact event occurred near the host rock of the nakhlite. Explaining the formation of symplectite and laihunite with the involvement of temporary warm water is difficult. This event was the source of all subsequent reactions as follows. Some of the other small craters are also distributed around the crater of the nakhlite ejection [8]. The timescale of alteration is unknown. However, the iddingsite formation could not have occurred within a short period because no brecciated or shock-molten areas were observed in Y 000593, although it experienced weak shock (5–14 GPa) [71].

2. Symplectite formation: The formation of symplectite in olivine indicates that the olivine was oxidized at high temperature, considered here as a shock-induced heat [9]. Symplectites lie parallel to the crystallographic axis (100) of the host olivine and are elongated along the [010] and [013] zones [9]. These symplectites were cross-cut by both sulfate-type iddingsite and Fe-Mg silicate after the siderite formation [16,72].

3. Crack formation: Cracks were physically formed in the nakhlite minerals including olivine by the impact event described above. Processes 1 to 3 might have occurred nearly simultaneously. The crack size could be related to the distance between each nakhlite position and the impact site.

4. Laihunite formation: Three-fold laihunite with a 3M structure forms along the walls of cracks at 800 °C to 400 °C [16]. Rapid oxidation occurred in olivine cracks by impact-induced hot gas (acid fog) at a short heating duration (e.g., <3 h) [56]. The degree of laihunite development increases closer to the impact site.

5. Post-impact hydrothermal fluid reaction: Iddingsite was formed as filling material in cracks by poorly crystalline materials occurring at a maximum temperature of 400 °C. In this phase, the detailed alteration process suggested in the present study is shown in the following processes, as extensively discussed in Section 4.4.

I. Oxidative fluid formed at the Martian surface, which caused alteration of the nakhlite host rock from the Martian surface to the subsurface zone. The melting of subsurface ice or permafrost by the heat originating from the shock event can be considered as the origin of this fluid.

II. Sulfate dissolution from FeS in mesostasis generated the oxidative and acidic fluid, which formed sulfate minerals such as jarosite.

III. The increase in pH by the aqueous alteration of silicate minerals caused the precipitation of FeOOH or the transformation of jarosite to FeOOH. MnOOH was also formed during this stage.

IV.     $CO_2$-rich reductive and alkaline fluid flowed from the subsurface ice by the residual heat. This fluid was incorporated into the nakhlite host rock along with iddingsite from bottom to top. The water reservoir would have been charged with $CO_2$ from the thicker Noachian atmosphere. Further discussion on the origin of this $CO_2$-rich fluid is given in Section 4.6.

V.     Carbonate was formed in the iddingsite. This fluid could not have reached the region that included laihunite because the coexistence of laihunite and siderite has not been reported. In addition, it is likely that not all of the sulfate veins were overprinted by the $HCO_3^-$-rich fluid because coexisting portions of sulfate and laihunite have been observed in Yamato 00 nakhlites [16,61].

VI.     Poorly crystalline clay minerals filled the remaining pores within the iddingsite.

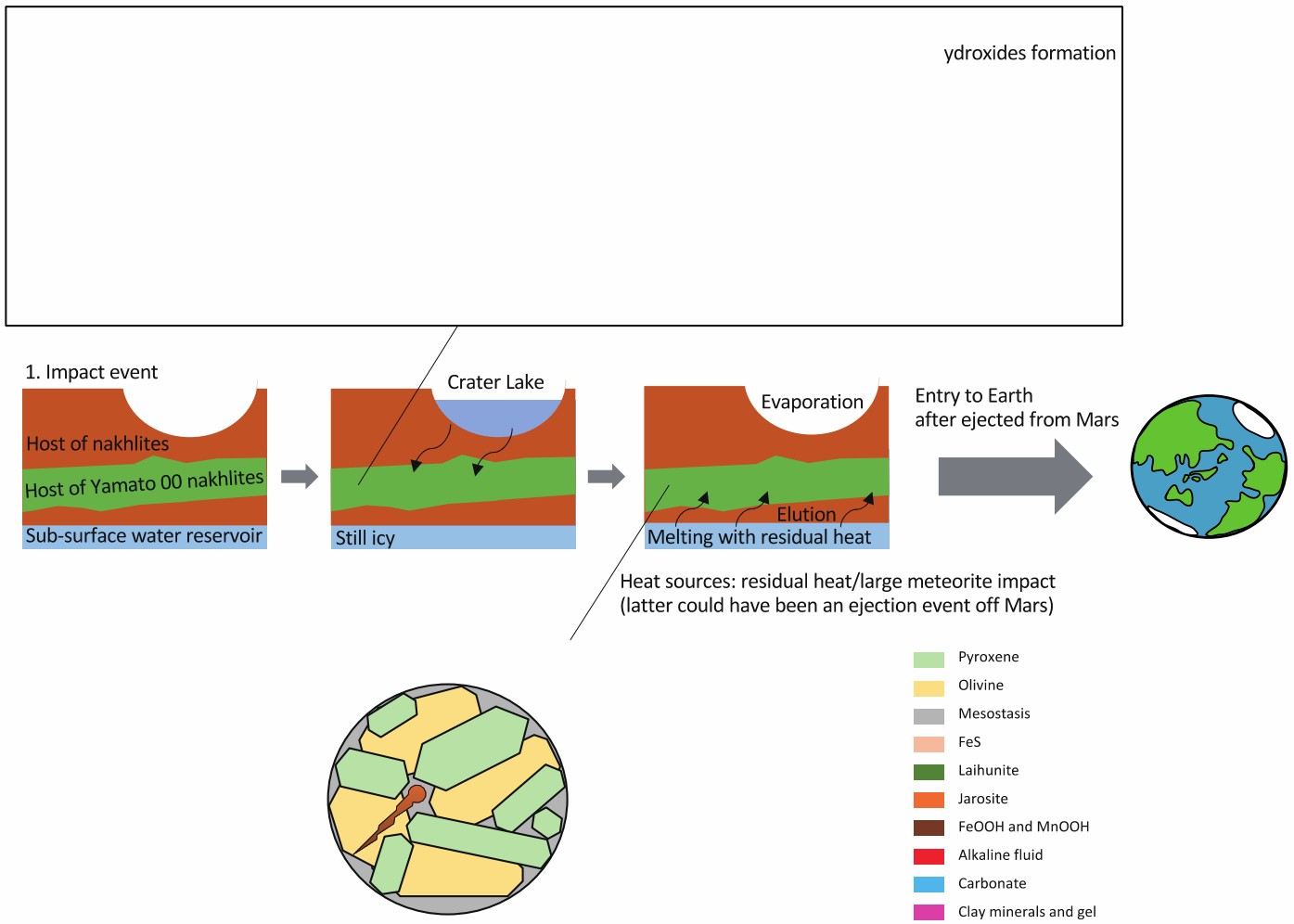

**Figure 10.** Ejection from Mars and journey to Earth.

### 4.6. Discussion of Consistency with Other Nakhlites and Pairs

Here, we compare Y 000593 and other nakhlites including its pairs to discuss their consistency, and we discuss a possible host rock of the Y 00 nakhlites. The serrated brittle fractures in the nakhlite filled in most cases by siderite are generally thought to be evidence of a nakhlite hydrothermal swarm created by impact adjacent to the nakhlite host rock (e.g., [15]). The origin of the $CO_2$-rich fluid is of particular importance.

Tomkinson et al. [17] clearly demonstrated that the coarsely serrated veins in Lafayette nakhlite were formed by crystallographically controlled carbonation of olivine. The initially penetrated vein of the ferroan saponite caused sequential $CO_2$ replacement reactions in olivine. The vein walls served as conduits for the $CO_2$-rich fluids to gain access

to the grain interiors. They suggested that no direct evidence is available to prove the siderite cementing of the serrated fractures is related to the shock-induced crack feature.

In Y 000593, however, areas are present with no serrated fractures, as shown in our PTS. In fact, it is difficult to explain whether the crack pathway formation of the oxic gases, which is related to the laihunite generation, was formed by crystallographically controlled siderite formation based on the formation sequence and the differences in formation temperature between the laihunite and siderite. Therefore, it is possible that some of the carbonate veins located in the deeper part, considered to be in the vicinity of Lafayette, in the host rock of Y 000593 were formed by crystallographically controlled carbonation of olivine by reductive alkaline fluid such as the siderite of Lafayette. This fluid could have been formed from a deeper part of the nakhlite body in Nakhla unreached by the impact-induced cracks related to the laihunite formation. This $CO_2$-rich fluid did not reach the laihunite-bearing upper part directly but was instead quickly consumed and exhausted in the lower part of the nakhlite host rock around Nakhla, GV, and Lafayette. Then, the remaining $CO_2$-depleted fluid penetrated to the upper part of the host rock with the dissolved surrounding minerals and slightly changed the solution composition. The stratigraphy of the nakhlite host rock producing carbonates and sulfates and the compositions of the clay minerals/gel showed an elemental ratio trend such that the Mg# became lower from bottom to top of the nakhlite position [24]. The elemental ratios in these clay minerals/gel could have a hidden connection to the type of mineral originally present in the fluid path upon penetration. Although previous studies have treated sulfate as terrestrial origin [15,24], the upper part of the nakhlite may have contained more elements originating from Martian sulfate such as S in its clay minerals/gel. In any case, it is considered that this neutral- to alkaline-pH, $CO_2$-rich fluid that formed the carbonate is representative of a major subsurface alteration fluid type in Martian impact terrains [25].

In previous studies, three formation mechanisms and three distribution models have been presented for each nakhlite in the host rock [9–12]. Our results support the Mikouchi formation model [10], in which the nakhlite host rock would have been formed as a single body. In addition, the subsequent alteration process described in the prescient study supports the Noguchi distribution model [16] of each meteorite based on the secondary mineral distribution.

A previous study confirmed the coexistence of clay minerals and laihunite in Y 000749 [62]. In addition, the iddingsite in Y 000749 contains poorly crystallized silica minerals and goethite as well as several sulfate grains [73], and some iddingsite of Y 000593 have similar mineral compositions [15]. Although Y 000593, 000749, and 000802 are considered as paired meteorites, the physicochemical environment of the iddingsites in these samples is different, which cannot be explained by differences in the local environment. The present study suggests that the host rock of Yamato 00 nakhlites might have been located at the layer between the region that generated oxidative acidic fluid and that producing reductive alkaline fluid. The main mass of the meteorite collected in NIPR (Y 000593, 29 × 22 × 16 cm, shown in the NIPR Meteorite Newsletter Vol. 10 No. 2) might have preserved the boundary area of the alteration process as described above.

*4.7. Discussion of the Origin of Jarosite and Other Sulfates in Y 000593*

Jarosite, which is clearly not formed by terrestrial weathering and did not occur as veins such as particulate, could be the result of the initial sulfate formation caused by primary oxidative acidic fluid. It is also possible that the jarosite was clogging the vein completely and did not allow carbonated water to flow where laihunite is prevalent. These veins might be interpreted as the sulfate-bearing iddingsite in Noguchi et al. [16], Suga et al. [61], and Shiraishi et al. [73].

In this study, we treated jarosite as the most easily explainable filling mineral in the primary sulfate-bearing vein based on the chemical species of the trace element speciations and their distribution processes. However, some could be gypsum depending on the cation environment in the fluid. We considered that clear sulfate veins related to

the fusion crust are terrestrial and likely overprint the Martian minerals. In contrast, we consider that the trace sulfate in the carbonate and the granular sulfate island adjacent to the clay minerals/gel are Martian sulfate that survived in the meteorite throughout the alteration events. It is easy to imagine that primary filled sulfates and Fe hydroxides were attacked by the penetration of fluid that formed clay minerals/gel and that some of the sulfate and Fe hydroxides could have remained as islands in the iddingsite. However, it is difficult to explain how these islands were formed by terrestrial contamination and not became the vein structure.

This remaining sulfate mystery will be solved by analyzing the hydrogen isotope ratios in the terrestrial jarosite vein, which is an obvious contaminant near the fusion crust [15,28], and in the patchy jarosite coexisting in the region between the clay minerals/gel and laihunite [15,73]. For such analysis, a local isotope analyzer such as NanoSIMS should be used because Martian sulfate likely remains in the interior of Y 000593.

## 5. Conclusions

In this study, we detected a carbonate phase in Y 000593 and characterized it as siderite and Mn-bearing siderite through FE-SEM-EDS/FE-EPMA prior to synchrotron microbeam X-ray microanalysis as a correlative microscopy experiment. Then, we applied synchrotron microbeam X-ray microanalyses including μ-XRF-XAFS and STXM to these carbonates in the iddingsite to reconstruct the formation environment of alteration minerals from the distribution and speciation analysis of several trace elements and to reconstruct the Eh-pH and temperature conditions of the alteration fluid involved in the formation of the iddingsite. These results show that two types of fluids were involved in the formation of the carbonate in the iddingsite and flowed in the order of oxidative acidic fluid followed by reductive alkaline fluid. In particular, the detection of Cr, a trace element component in Martian meteorites, was examined via SR-μ-XRF-XAFS analysis using synchrotron radiation facilities to identify the chemical species. In addition to the interpretation from conventional geochemical laboratory experiments, trace amounts of Cr(III) left in the meteorite confirmed the existence of $MnOOH/Mn_2O_3$ in the past as a precursor and constrained the pH conditions of the liquid phase at the time of Cr(III) incorporation. Although we need to investigate the isotope feature of the water preserved in the clay minerals or sulfates, the present study provides unique information based on trace element geochemistry to solve the question of the coexistence of carbonates and sulfates in nakhlites iddingsite, which has not been solved in the literature. Herein, tracing the fluid chemistry and condition provided a new constraint on the physicochemical evolution of the water that altered the nakhlite body during the last ~6.5 million years.

**Supplementary Materials:** The following are available online at www.mdpi.com/2075-163X/11/5/514/s1, Figure S1: Carbonate analysis points plotted on backscattered election (BSE) images, Figure S2: Representative Mn K-edge XANES spectra (pink spectra) obtained from the Mn-bearing carbonate iddingsite, Figure S3: Analysis points of iron sulfides plotted on the optical microscopic images of PTS 58-1 (left) and 63-3 (right) with S-XANES spectra, Figure S4: Analysis points of Cr-XANES indicated over the BSE, RPMA, and μ-XRF images, Table S1: Characteristics of the beamlines used in this study with information of the elements measured, Table S2: Carbonate composition in iddingsite measured via electron microprobe analysis (wt.%). The name corresponds to the analysis point in Figure S1, Table S3: Sulfate content in iddingsite measured via electron microprobe analysis (wt.%).

**Author Contributions:** H.S. and K.S. conceptualized and designed the research; Y.T. (Yoshio Takahashi) supervised the research; H.S., T.U., and Y.T. (Yoshio Takahashi) conducted project administration. H.S., K.S., and Y.T. (Yoshio Takahashi) performed research, analysis, investigation, data curation, data analysis, and wrote original draft; A.Y. provided samples; O.S., K.N., Y.T. (Yasuo Takeichi), and T.O. provided synchrotron investigation methods; and H.S., K.S., T.U., A.Y., T.O., and Y.T. (Yoshio Takahashi) edited the final version of this paper. All authors have read and agreed to the published version of the manuscript.

**Funding:** This work was funded by a Grant-in-Aid for Scientific Research (JSPS KAKENHI Grant Nos. 17H06454, 17H06458, 18H04134, 19K14780, and 19H01960). Some parts of the technical method and idea were obtained in the international scientific research collaboration, which was supported by JSPS Core-to-Core Program "International Network of Planetary Sciences".

**Data Availability Statement:** The data presented in this study are available from the corresponding author, H.S., upon reasonable request.

**Acknowledgments:** We are grateful to Masaaki Miyahara for the significant and comprehensive discussion on the entire study and for acquiring the UVSOR machine time. We are also thankful to Natsumi Sago and Naoki Shiraishi for their significant discussion on parts of this study. Some parts of the technical method and idea of this study were obtained through an international scientific research collaboration, supported by the JSPS Core-to-Core Program "International Network of Planetary Sciences." We would also like to thank Daisuke Wakabayashi and Shohei Yamashita for their support regarding the STXM measurement at BL-19A, along with Hayato Yuzawa for providing STXM support at UVSOR BL4U. We appreciate Hideto Yoshida and Koji Ichimura for offering revision advice, lectures, and significant support for the FE-SEM and FE-EPMA investigations performed at U-Tokyo and Masahiro Yasutake at JASRI for the support on carbonate calculation of EPMA data for me. We are grateful to Mr. Rei Kanemaru for the FE-SEM analysis support at NIPR and to Arisa Usui and Kanta Ono at KEK-PF for the adjustment and management of FIB, respectively. The synchrotron radiation experiments were performed at the BL37XU of SPring-8 with the approval of the Japan Synchrotron Radiation Research Institute (JASRI) (proposal nos. 2018A0148, 2019A1523, 2019B1498, and 2020A0174). This work was performed under the approval of the KEK Photon Factory Program Advisory Committee (proposal no. 2018S1-001). Part of this work was performed at the BL4U of UVSOR Synchrotron Facility, Inst. Mol. Sci. Okazaki (IMS program no. 2019A-543). We are grateful to all of the PF, UVSOR, and JASRI/SPring-8 staff for their support and to Tomoko Ojima at NIPR for the curation and distribution of Y 000593 samples (proposal nos. 1725 and 1732).

**Conflicts of Interest:** The authors declare no conflict of interest.

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
