# Peer review of "A New Constraint on the Physicochemical Condition of Mars Surface during the Amazonian Epoch Based on Chemical Speciation for Secondary Minerals in Martian Nakhlites"

_minerals, doi:10.3390/min11050514_

Round 1

Reviewer 1 Report

Dear Professor Suga and co-authors,

Please fine my comments and suggestions for the manuscript ‘A new constraint on the physico-chemical condition of Mars 2 surface during Amazonian epoch based on chemical speciation 3 for secondary minerals in Martian nakhlite meteorite body’.  This manuscript presents an interesting multi-modal approach to address a paradox in the formation of secondary minerals in the Y000593 nakhlite meteorite. Specifically, they use a combination of electron microanalysis and synchrotron x-ray microscopy techniques to unravel how both carbonates form in alkaline conditions can be found to co-exist with sulfates forms under acidic conditions. As physical samples of Martian materials are limited due to only what has been transported to Earth via ancient impact events these samples require the extensive and detailed microscale analysis presented here. Overall, the experiments here piece together a convincing model for the Martian processes which could have given rise to these mineral textures observed.

The manuscript in its current form, however, does need some structural attention to improve the clarity and understandability of the series of experiments presented to be suitable for publication. My major recommendations would be as follows:

  • (line 19-20) - The opening sentence of the Abstract does not read clearly. Are you referring only to the secondary minerals in iddingsite? Implying that all the secondary minerals are iddingsite? Please re-word to clarify that this is looking at only the secondary minerals found in the iddingsite in the meteorite.
  • (line 23-28) First I would add ‘meteorite’ before the sample name Y000593. Second and more significantly this sentence descries how the experiment was conducted noting the use of electron microscopy techniques (EDS and EPMA) as well as the synchrotron technique of µ-XRF–XAFS.
    1. Both the introduction and conclusion do not mention the use of the electron microscopy techniques. Whereas when you read the paper these techniques were used in a correlative manner to guide the synchrotron experiments. It would be more accurate to describe this as a correlative microscopy experiment.
    2. Secondly, µ-XRF–XAFS, is not the only synchrotron x-ray microscopy technique used. It would make more sense to state that the project used multiple synchrotron techniques including µ-XRF–XAFS.
  • (line 147 -153) fig 1 and optical microscopy.
    1. First the ‘reddish vein of iddingsite’ is not clear in either Fig 1a or Fig 1b? Would it be possible to label?
    2. Were both thin sections used? If so what are the differences between the two?
    3. Fig 1b needs a scale bar, unless it is at the same scale as 1a, but then it needs to be stated.
    4. Can you crop out more of the epoxy mounting material and fill available layout space with more grain information? Most of the grain details in these images are impossible to read.
    5. Is it possible to label on these images where the following SEM and µ-XRF–XAFS experiments were performed? All the experiments are built up from features seen in these images, but the reader is not able to place the measurements in the context of this optical information or more specifically how they relate to one another spatially.
  • (line 209) I am not clear why this table is here. While useful information it does not drive the story forward. I would recommend that this go into a supplemental materials document, along with what seems to be much of the unpublished but alluded to data from these techniques.
  • I would prefer to see the results and discussion spit into two sections. While I do see the purpose of writing these as a single section, at times discussion of results from other reports clouds the new results published here.
  • (line 272-2283) Figure 2 requires effort to turn into a consistent and informative figure.
    1. First, I would prefer to see the BSE image placed in the upper left corner, so that when looking at the element maps, I have a rough idea of the microstructure.
    2. Is it possible to replot both of these figures (2a and 2b) using a uniform colour system either both in Rainbow (or better a perceptually uniform colour scale like ‘viridis’). It is difficult to make qualitative comparisons between the two sets of maps.
    3. Fig 2a, for the colormaps included, is there a scale like background corrected counts? Even just labelling high to low would be useful.
    4. Fig 2b, The C, Ca, and Mn images are just black image. If there is data in those images, then adjusting the gamma / using autolevels etc to bring out the low level information (and noting this) would be appropriate. Further, for these images are their scales to the intensity of the data collected? Even absolute counts would be appropriate?  Especially the Ca map, as it looks like it is largely just mapping the background.
    5. All the issues with Fig 2 could be addressed by using post-processing code designed for the task like Hyperspy (open-source Python 3 package), this would allow for background correction and then plotting of element maps using a uniform scale.
  • (lines 284 -288) Fig 3 presents results of spot analysis across the sample(s?), however, I am unclear how these measurements spatially relate to the information presented in all the other figures.
    1. Does a map of where these measurements were done exist?
    2. I am unclear why the XANES reference spectra are presented in this figure. It makes the direct reference in the text (line 256 -258) confusing since that section only discuss the Mn content. Not how it was derived and compared against these reference spectra.
  • (lines 300-303) I am not clear if you are trying to argue that there is no contamination in the sample. This is one of the points where the discussion and the results are getting confused and used to cross purposes.
  • (Line 317) Refers to Fig 5 a-c as ‘SEM–EDS, EPMA, and µ-XRF–XAFS’, however the caption for 5c clearly states that this is a STXM. Is this a typographic mistake, or are the two techniques the same? If they are the same please pick one term and use throughout the manuscript.
  • (Line 334) Fig 5b – how are these spectra pulled out of the EDS map presented in Fig 5a? are they single point spectra (if so show where)? Or are they regions drawn? Again, by separating the discussion from the results you would have space to discuss how these spectra were obtained. Right now, I do not have information on how to repeat these experiments on another sample if I wanted to.
  • (lines 410-417) Figure 7 this is the only µ-XRF–XAFS data presented in the entire paper, which is fine, but the introduction, abstract and conclusion all make it seem like there should be more, or at least this should seem more significant to the conclusions of the story. Further, I am not clear why there are more XANES reference spectra in a figure about µ-XRF–XAFS.
  • (lines 650) after step 5VI am I right in understanding that there is a latter second impact event which launched this sample onto it’s eventual journey to Earth? If so, I would maybe suggest for clarity changing the caption in Fig 10 to ‘Ejection from Mars and Journey to Earth’ as it orders the events into the correct chronological order.

I hope that this comments are helpful in revising this exciting manuscript.

regards

Reviewer 2 Report

Edit the title to e.g. '...in the Martian nakhlite'.

Line 45 - give the chronometer for this age.

Line 51 Reference for age and nakhlite formation needed.

Line 51 needs rewriting as clearly not all volcanism in Tharsis and the northern lowlands originated with O. Mons.

Line 58 - needs some rewriting: the Chassignite (dunite) meteorite parent rocks. Relevance to this paper?

Line 95. Not entirely true e.g. see Bridges and Schwenzer 2012 EPSL for a discussion of fluid alteration conditions. Nakhla carbonate (Fe-Mg siderite) is consistent with 75–100 °C, pH 4, W/R 100, with saponite and serpentine-like phases that partially replace it forming at lower T's By this model and also Bridges and Grady 2000 EPSL, the sulphate is a late stage product of this brine, evaporating from the final fluid.

Line 119. I fear the simplest explanation for the juxtaposition of jarosite and siderite is simply that the former is terrestrial.  The terrestrial origin of jarosite in Y000749 is clear e.g. see Changela et al. 2010 and references therein.  Y000593 is less weathered than Y000749 but the Antarctic link makes a similar origin for jarosite in Y000593 a distinct possibility.

Line 137.  Not true really, Y000749 is heavily altered relative to the other nakhlites.

Line 1444. The amorphous gel is clay-like in composition - ferric saponite (Hicks et al. 2014) - its not amorphous SiO2. Opaline silica is in very minor abundances relative to the clay-gel.

A recent review of the carbonate in the nakhlites is given in Bridges et al. (2019) Carbonates on Mars. In ‘Volatiles on Mars’. 

Line 258 - previous (first peer reviewed) siderite and Ca-siderite/ankerite compositions for the nakhlites should be shown for comparison e.g.  summarised with references therein by Bridges et al. 2019.

Line 260 - see point above about references to previous analyses of siderite (including Mn-rich) in the nakhlites.  

However, a more serous issue with the current manuscript with regard to the siderite (and sulphate) analyses is that there is no table of data. The identification of significant amounts of preserved siderite in Y000593 would certainly be noteworthy but the probe data needs to be shown. Similarly, I cant see any discrete siderite grains highlighted in the BSE and X-ray figures. There are high Mn-Fe patches in Fig. 2 which might  be relict siderite but the BSE image isn't really good enough to show if it is a discrete carbonate grain.

Without that that basis and context, the interpretation of the XAS analyses and the synthesis discussion about the nakhlites have limited strength in the manuscript.

Line 422 - see point above about needing a table of data.

From my own studies of Y000593 I suspect the authors have identified heavily altered/replaced siderite.  That doesn't preclude publication but the compositions do need to be clearly shown and explained.

Round 2

Reviewer 1 Report

Dear Professor Suga and co-authors,

I am pleased to recommend the revised manuscript ‘A new constraint on the physico-chemical condition of Mars 2 surface during Amazonian epoch based on chemical speciation 3 for secondary minerals in Martian nakhlite meteorite body’ for publication.  There have been significant changes to the manuscript both in response to my comments and the other reviewer’s resulting in an overall clearer message about the phases measured and how they relate to the mineral evolution processes which they experienced on Mars.

I would say that I still have found a few minor points that I would suggest be addressed before final publication.

  • There are still several typographic errors. A careful proof reading by a third party would push the quality of this manuscript to a higher level. For example, on line 133 it should read 'juxtaposition' not 'just a position'.
  • The phrasing of lines 225 and 234 relating to the pixel size and areas scanned using synchrotron techniques is a little odd. For example, I would re-write line 225 to read: ‘…μ-XRF imaging analysis was performed with a step width of 0.1 μm per pixel, with an image area of 400 × 400 pixels...’

Finally, I find the revision of Fig 2 acceptable, especially as using a uniform colour scale makes it easier to compare the elements more directly.  However, this still does not fully address my comment. Even with out the use of standards, EDS is quantitative, meaning one should be able to compare the high’s and low’s in a relative manner, say by scaling the total intensity of each element map by the total counts of the map.  The way these images stand now, I still do not know how to compare the Si map high (Fig 2a) with the Ca high. Are these the same number of counts? While I would like to see a more quantitative presentation of the data, the current presentation seems to fit with the accepted standards of the planetary community and so would not use this comment as a barrier to publication. I would however encourage that in future studies take a more quantitative approach to EDS and synchrotron data analysis.

I look forward to seeing this manuscript being published and believe it will contribute to extending our understanding of the geological processes active on an ancient Mars.

Reviewer 2 Report

Suga et al. have made substantial and positive changes to the manuscript since the first round of reviews.  In particular there is now a credible table of carbonate probe data. This could still be improved with information about whether the data were recalculated SiO2-free for instance, and some saponite and sulfate analyses. However, I suggest that is now at the discretion of the authors basically about how widely used they hope the paper will be.

I dont think the authors have really established the early, pre-clay and siderite formation of jarosite in Y000593 in their preferred model (does any clay cross-cut jarosite?) and it may well be terrestrial, but readers of a published manuscript can make their own decisions about that and it need not stop publication in Minerals as there is enough solid data about this important Mars sample in the paper.

The Cr and Mn XAS data is a useful addition to the literature. See my comments on the attached pdf about some edits and clarification needed before publication.

Other comments, corrections and suggestions on the annotated pdf.
